# SCHEMAS: Leveraging Scalable Heterogeneous Graph for Query-Guided Reasoning in Multi-Agent Systems

## Abstract

Multi-agent systems have demonstrated strong problem-solving capabilities, yet real-world tasks often require complex interactions that involve retrieving external knowledge and calling tools. In such settings, interaction dependencies among query, agent, and tool entities are often encoded implicitly and transiently in dialogues and execution traces, without an explicit structure to systematically represent invocation conditions and collaboration pathways. In this paper, we propose SCHEMAS, a scalable and configurable heterogeneous graph framework for multi-agent systems that unifies four entity types: query, agent, tool, and distilled insight, explicitly capturing their typed interactions to enable query-guided reasoning and continual evolution. To improve performance on a new query, we propose a query-driven structured memory mechanism that constructs a memory subgraph by retrieving the most relevant query, agent, tool, and insight nodes from the heterogeneous graph. Based on the memory subgraph, we allocate the query to the most compatible agent by matching the query's required capability profile to agent capability representations. After execution, we apply the dynamic update mechanism that writes back the new query and its trace to the heterogeneous graph, thereby enabling the progressive evolution of the multi-agent systems. Extensive experiments across eight benchmarks spanning tool-use and memory-augmented interactive settings demonstrate that SCHEMAS delivers consistently strong performance, scales across diverse tool-use scenarios and interactive tasks, and does so with **lower token usage** than state-of-the-art baselines.

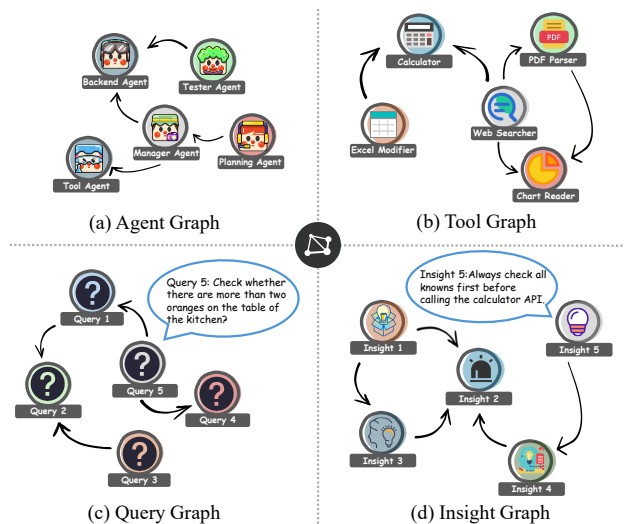

*Figure 1.* The overview of SCHEMAS's subgraph, comprising the agent graph, tool graph, query graph, and insight graph.

[1]Anonymous Institution, Anonymous City, Anonymous Region, Anonymous Country. Correspondence to: Anonymous Author <anon.email@domain.com>.

## 1. Introduction

Large language model (LLM)-driven multi-agent systems (MAS), which assign distinct roles and capabilities to multiple agents and enable their interactions, have been applied across a wide range of domains, including embodied tasks (Wang et al., 2023a; Li et al., 2024a), code generation (Wu et al., 2024a; Guo et al., 2024a), and data analysis (Hong et al., 2025; Guo et al., 2024b). However, despite this progress, current MAS still struggle with reliable long-horizon planning, long-term memory, robust tool management, and effective coordination (Erdogan et al., 2025; Hatalis et al., 2023; Dong et al., 2025; Sun et al., 2025). These challenges are further amplified as MAS scale in the number of agents, the diversity of tools, and the variability of tasks.

To address these challenges, researchers have increasingly turned to graphs as as a powerful auxiliary structure to organize, enhance, and interpret the execution flows of LLM agent systems (Zhuge et al., 2024; Zhang et al., 2024a). Unlike the sequential language data on which LLMs are grounded, graphs are expressive and general-purpose data structures that naturally encode complex relationships

among entities, query, tool, and agent (Liu et al., 2025; Bei et al., 2025). Recent works have leveraged this capability by modeling various components of multi-agent systems as graphs, such as tool graph (Liu et al., 2024b;a; Ding et al., 2025; Wang et al., 2025), agent graph (Qian et al., 2024; Zhang et al., 2025b; 2024c;b), task graph (Zhang et al., 2025d; Wu et al., 2024b), environment graph (Huang et al., 2025; Chen et al., 2025), and insight graph (Zhang et al., 2025a). However, these graphs are often constructed around a single subsystem (e.g., tools, agents), which yields fragmented structural priors and makes it difficult to capture cross-component dependencies.

Heterogeneous graph provides a natural representation for modeling MAS to capture cross-component dependencies, where different types of nodes and relationships can be explicitly modeled (Zhang et al., 2025c). D$^3$MAS (Zhang et al., 2025c) constructs a heterogeneous graph to support knowledge sharing in MAS via a decompose–deduce–distribute pipeline and another work employs heterogeneous graph memories to distill reusable strategies from agent trajectories (Xia et al., 2025). However, existing heterogeneous graph designs are largely objective-specific and rarely turn dialogues and execution traces into an explicit, reusable interface for coordination, continual updates, and self-evolution.

**Our proposal**. Building on the above observations, we propose SCHEMAS, a scalable heterogeneous graph framework that unifies four entity types-query, agent, tool, and distilled insight-under a single, extensible schema. Figure 1 visualizes these structures, and their formal definitions are placed in Section 3. When a new query arrives, we first perform query-driven structured memory retrieval in a coarse-to-fine manner: (i) we retrieve semantically similar historical queries via kernelized embedding similarity, and (ii) we refine the candidates using meta-path consistency, before expanding to their associated agents, tools, and distilled insights to assemble a structured memory subgraph for the incoming query. Based on this memory subgraph, we match the query's required capability vector to agent capability vector to select the most compatible agent and choose an appropriate execution strategy (*Direct*, *Decompose* or *Collaborative*). After execution, we perform trace-driven heterogeneous graph updates by inserting the new query, revising node states, reweighting collaboration and tool use relations, and integrating distilled insights to enable continual self-evolution. This closed-loop design turns long-horizon interaction traces into structured, reusable experience, allowing the system to progressively improve retrieval quality, agent assignment, and tool orchestration as more tasks are solved. Extensive experiments on diverse benchmarks demonstrate that SCHEMAS consistently outperforms strong baselines while reducing interaction cost.

**Contributions**. Our main contributions include:

**1. Scalable Heterogeneous Graph:** We propose a novel framework for MAS that leverages a unified, scalable heterogeneous graph to explicitly model the interactions between tool, query, agent, and insight.

**2. Structured Memory Mechanism:** We propose a query-driven structured memory mechanism that constructs a subgraph for efficient retrieval and decision-making, improving efficiency and coordination on new queries.

**3. Dynamic Update Mechanism:** We propose a trace-driven dynamic update mechanism that writes back each new query and its execution trace to the heterogeneous graph, enabling continual self-evolution.

**4. Superior Performance:** Our experiments demonstrate that the SCHEMAS consistently improves both accuracy and efficiency across diverse tasks, achieving lower token usage than the state-of-the-art methods.

## 2. Related Work

### 2.1. Graph-Augmented Multi-Agent Systems

Recently, researchers have increasingly turned to graphs as a complementary infrastructure to organize, enhance, and interpret the execution flows of LLM agent systems (Zhuge et al., 2024; Zhang et al., 2024a). Recent works have leveraged this capability by modeling various components of multi-agent systems as graphs (see Fig. 1 and Fig. 4), such as tool graphs (Liu et al., 2024b;a; Ding et al., 2025; Wang et al., 2025), agent graphs (Qian et al., 2024; Zhang et al., 2025b; 2024c;b), task graphs (Zhang et al., 2025d; Wu et al., 2024b), workflow graphs (Wei et al., 2025), environment graphs (Huang et al., 2025; Chen et al., 2025), knowledge graphs (Jiang et al., 2025; Zhou et al., 2024) and insight graphs (Zhang et al., 2025a). Additionally, some works have preliminarily explored the use of heterogeneous graph to capture more intricate relationships between different entities (Zhang et al., 2025c; Xia et al., 2025). *Difference compared to existing work*: While prior graph-augmented MAS methods typically model a single component (e.g., tools or agents) or adopt objective-specific heterogeneous graph, SCHEMAS provides a unified and extensible heterogeneous graph that explicitly models queries, agents, tools, and insights, thereby enabling continual trace-driven refinement within MAS.

### 2.2. Structured Memory in Multi-Agent Systems

Recently, researchers have proposed different types of structured memory aimed at organizing and optimizing information storage and retrieval through explicit topological structures (Hu et al., 2025b). Two main types of structured memory include Planar (2D) Memory and Hierarchical (3D)

Memory. Planar (2D) Memory organizes memory units within a single structural layer, providing a planar memory management approach. Its topological structure can be a graph (Xu et al., 2025; Kim et al., 2025; Yuen et al., 2025; Long et al., 2025), tree (Aadhithya et al., 2024; Rezazadeh et al., 2024), or hybrid structure (Li et al., 2024b; Lei et al., 2025). Hierarchical (3D) Memory organizes information across layers, using inter-level connections to shape the memories into a volumetric structured space. Research has explored pyramid-based memory structures, where information is progressively organized into higher layers of abstraction and queried in a coarse-to-fine manner (Edge et al., 2024; Hu et al., 2025a; Zhang et al., 2025a; Rasmussen et al., 2025; Wang et al., 2024b). In contrast, multi-layer memory structures emphasize layered specialization, organizing memory into distinct modules or levels, each focusing on specific information types or functions (Anokhin et al., 2024; Wu et al., 2025; Li et al., 2025a; Jimenez Gutierrez et al., 2024; Sun & Zeng, 2025). In particular, G-Memory (Zhang et al., 2025a) is closely related to SCHEMAS, as it organizes MAS experience into a hierarchy of graphs and retrieves reusable collaboration patterns via cross-level traversal; however, it does not model all MAS components as entities within a single unified heterogeneous graph with explicit typed relations. *Difference compared to existing work*: Inspired by G-Memory, we leverage a unified heterogeneous structure to make cross-component dependencies explicit and operable, enabling compositional subgraph retrieval. In contrast, hierarchical multi-graph memories typically depend on cross-layer indexing and traversal, making unified reuse and refinement across components less direct.

## 3. Preliminary

Formally, we define a directed heterogeneous graph as $\mathcal{G} = (\mathcal{V}, \mathcal{E}, \mathbb{V}, \mathbb{E}, \varphi, \psi)$ where $\mathcal{V}$ is the set of concrete entities (nodes) in the system, each representing an instance such as an individual agent, a tool module. $\mathcal{E} \subseteq \mathcal{V} \times \mathcal{V}$ is the set of directed edges, where each edge $(u, v)$ encodes dependencies or information flow between entities. $\mathbb{V} = \{\texttt{Agent}, \texttt{Tool}, \texttt{Query}, \texttt{Insight}\}$ is the node-type space describing the categories of entities. $\mathbb{E} = \{\texttt{communicates}, \texttt{calls}, \texttt{assigns}, \texttt{consumes}\}$ is the edge-type space describing interaction types among entities. The mapping functions $\varphi : \mathcal{V} \to \mathbb{V}$ and $\psi : \mathcal{E} \to \mathbb{E}$ assign each node and edge to specific types from sets. Specifically, the node and edge types listed are just a subset of the many possible types that can be included, and the graph structure is designed to be extensible to accommodate additional types as needed.

**Definition 3.1** (Meta-path). Given a heterogeneous graph $\mathcal{G} = (\mathcal{V}, \mathcal{E}, \mathbb{V}, \mathbb{E}, \varphi, \psi)$, a *meta-path* $P$ is a typed path pattern specified by an alternating sequence of node types and relation types:

$$P : A_1 \xrightarrow{R_1} A_2 \xrightarrow{R_2} \cdots \xrightarrow{R_l} A_l,$$

where $A_i \in \mathbb{V}$, $R_i \in \mathbb{E}$, $\forall i \in [l]$. Meta-paths capture higher-order semantic dependencies across heterogeneous entity types. For example, $\texttt{Query} \xrightarrow{\texttt{assigns}} \texttt{Agent} \xrightarrow{\texttt{calls}} \texttt{Tool}$ represents a query being executed by an agent via tool invocation.

**Heterogeneous Graph Architecture.** Our framework organizes agents, tasks, tools, and distilled insights into a unified heterogeneous graph through four different subgraphs:

**Query Graph.** The query graph captures historical tasks and their metadata, which is the subgraph of the global heterogeneous graph $\mathcal{G}$ restricted to query-type nodes. Specifically, we define the query node set as $\mathcal{Q} = \{v \in \mathcal{V} : \varphi(v) = \texttt{Query}\}$, yielding the query graph $\mathcal{G}_{\text{query}}$:

$$\mathcal{G}_{\text{query}} := (\mathcal{Q}, \mathcal{E}_{\text{query}}) = \left(\{(Q_i, C_i^{\mathcal{Q}}, \Phi_i)\}_{i=1}^{|\mathcal{Q}|}, \mathcal{E}_{\text{query}}\right), \quad (1)$$

where each node $q_i := (Q_i, C_i^{\mathcal{Q}}, \Phi_i)$ consists of the query $Q_i$, the required capability vector $C_i^{\mathcal{Q}}$ for the query, and the execution status $\Phi_i \in \{\texttt{Succeeded}, \texttt{Failed}\}$. Edges $\mathcal{E}_{\text{query}} \subseteq \mathcal{Q} \times \mathcal{Q}$ encode semantic similarity across tasks and support efficient retrieval of relevant prior experiences.

**Agent Graph.** We define the agent graph as a type-induced subgraph of the global heterogeneous graph $\mathcal{G}$ that models agent-specific behaviors and interactions during task execution:

$$\mathcal{G}_{\text{agent}} := (\mathcal{A}, \mathcal{E}_{\text{agent}}) = \left(\{(A_i, C_i^{\mathcal{A}}, \Psi_i)\}_{i=1}^{|\mathcal{A}|}, \mathcal{E}_{\text{agent}}\right). \quad (2)$$

Each agent node is represented as $a_i := (A_i, C_i^{\mathcal{A}}, \Psi_i)$, where $A_i$ represents the underlying LLM instance, $C_i^{\mathcal{A}}$ is the agent capability vector, and $\Psi_i \in \{\texttt{Idle}, \texttt{Acting}, \texttt{Waiting}, \texttt{Done}, \texttt{Error}\}$ denotes the agent's current state. Edges $\mathcal{E}_{\text{agent}} = \{(a_i, a_j) : a_i, a_j \in \mathcal{A} \wedge \psi(a_i, a_j) \in \{\texttt{communicates}, \texttt{assigns}\}$ represent interactions or dependencies between agents, including communication and assignment. These edges are weighted by a matrix $W^{\mathcal{A}}$, which defines the strength of dependency between agents. Specifically, $W^{\mathcal{A}}(i, j)$ quantifies the degree of coordination or dependency between agent $A_i$ and agent $A_j$, based on their collaborative success, proximity in task execution, and ability alignment. This graph provides information for downstream analysis such as task allocation, performance assessment, and efficiency optimization.

**Tool Graph.** The tool graph models functional tools, APIs, and their compositional structure:

$$\mathcal{G}_{\text{tool}} := (\mathcal{T}, \mathcal{E}_{\text{tool}}) = \left(\{(T_i, \xi_i, \zeta_i)\}_{j=1}^{|\mathcal{T}|}, \mathcal{E}_{\text{tool}}\right), \quad (3)$$

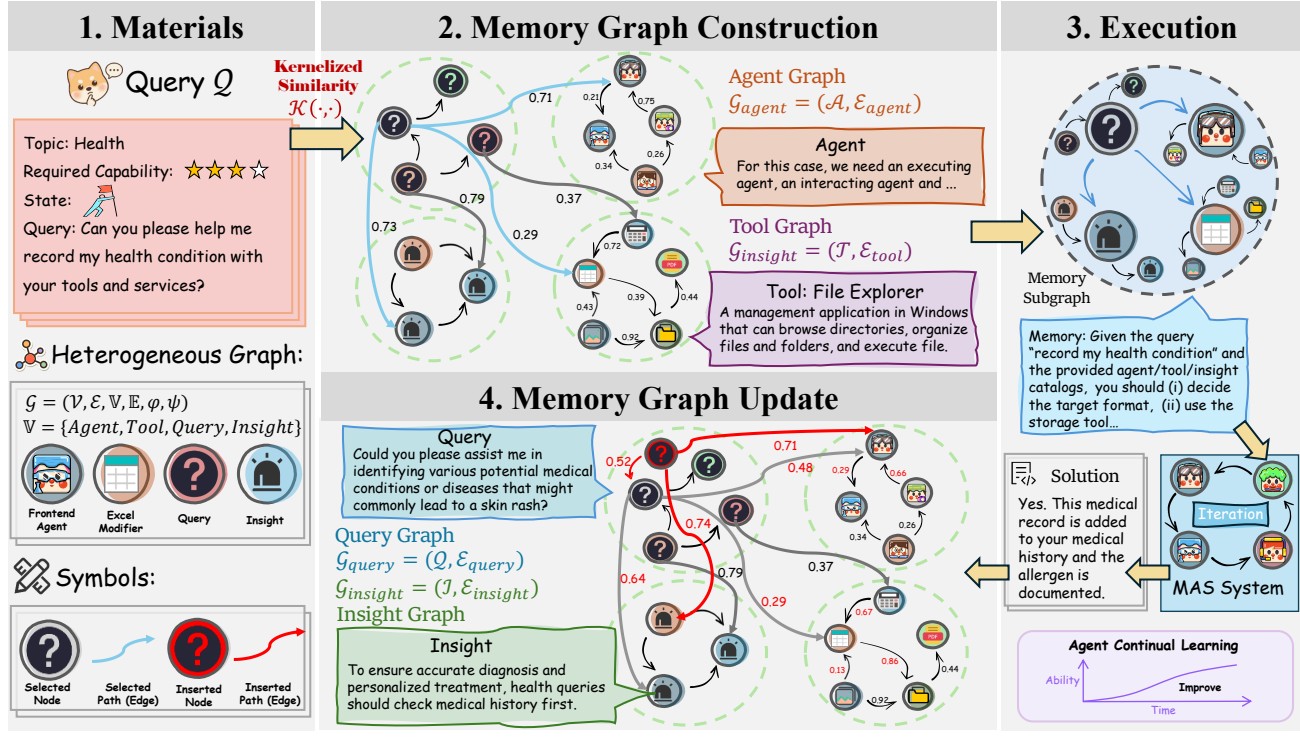

*Figure 2.* The overview of our proposed SCHEMAS. For a new query , it (1) constructs a query-conditioned memory subgraph via coarse-to-fine query retrieval and trace-neighborhood extraction, (2) routes the query to a compatible agent through capability matching, and (3) writes back the resulting execution trace to update the graph for continual refinement.

where each tool node $t_i$ is a tuple $(T_i, \xi_i, \zeta_i)$. Here $T_i$ denotes the tool identity, $\xi_i$ denotes static tool descriptions (e.g., tool description and input/output schemas), and $\zeta_i$ denotes dynamic tool statistics (e.g., usage count, success rate, and cost) that can be updated online from execution traces. Edges $\mathcal{E}_{tool} \subseteq \mathcal{T} \times \mathcal{T}$ encode compositional or substitutable relations such as schema compatibility, prerequisite constraints, or functional equivalence. A weighted adjacency matrix $W^{\mathcal{T}}$ is maintained to quantify the strength of tool composition, where $W^{\mathcal{T}}(i, j)$ weights the connection from $t_i$ to $t_j$. This graph enables structured tool selection and multi-step tool planning.

**Insight Graph.** High-level reusable strategies distilled from multiple task trajectories form the insight graph:

$$\mathcal{G}_{insight} := (\mathcal{I}, \mathcal{E}_{insight}) = \Big(\{(\kappa_i, \Omega_i)\}_{i=1}^{|\mathcal{I}|}, \mathcal{E}_{insight}\Big), \quad (4)$$

where each insight node $\iota_i := (\kappa_i, \Omega_i)$ contains distilled strategy content $\kappa_i$ and its supporting set $\Omega_i \subseteq \mathcal{Q}$ that records the historical queries whose trajectories give rise to this insight. Edges $\mathcal{E}_{insight} \subseteq \mathcal{I} \times \mathcal{I} \times \mathcal{Q}$ form query-conditioned hyper-relations, where $(\iota_i, \iota_j, q)$ indicates that insight $\iota_i$ contextualizes (e.g., refines or specializes) insight $\iota_j$ under query $q$ (Zhang et al., 2025a). This graph expresses transferable reasoning, tool use patterns, and agent collaboration experiences.

# 4. Query-Guided Structured Memory Graph Construction

In this section, we describe how SCHEMAS constructs a query-guided structured memory graph for an incoming query $Q_{new}$. As illustrated in Figure 2, upon the arrival of $Q_{new}$, SCHEMAS performs coarse-to-fine retrieval to identify relevant historical queries and enforce schema-level consistency via structural meta-paths. It then builds a memory subgraph by aggregating $h$-hop trace neighborhoods anchored at the retrieved queries, and routes $Q_{new}$ to a compatible agent via capability matching for execution.

**Query Selection based on Structural Meta-path.** When a new query $Q_{new}$ arrives, the system first performs a coarse-grained retrieval over the query graph. We decouple representation from similarity measurement by adopting an encoder $f_\theta(\cdot)$ together with a kernelized similarity function $\mathcal{K}(\cdot, \cdot)$, such as a cosine kernel, an RBF kernel, or a Gaussian kernel. Concretely, we map each query into a fixed-length embedding $z(Q_{new}) := f_\theta(Q_{new})$ and rank historical queries $Q_i$ by the kernel score $\mathcal{K}(z(Q_{new}), z(Q_i))$, yielding the top-$k$ candidate set:

$$\mathcal{Q}_{new}^S = \arg \text{top-}k_{q_i \in \mathcal{Q}} \, \mathcal{K}\big(z(Q_{new}), z(Q_i)\big).$$

This query selection strategy strengthens coarse retrieval

by combining richer semantic representations with a more expressive similarity metric, thereby enabling more flexible and query-sensitive retrieval while preserving the same coarse-to-fine pipeline.

However, embedding proximity mainly captures semantic resemblance and may overlook whether two tasks exhibit consistent behaviors. Since $Q_{\text{new}}$ has not been executed yet, it lacks concrete interaction edges in the memory graph. We thus obtain an intent sketch (e.g., GPT-4o) from a LLM (see prompt in Appendix C), which predicts a tool set $\widehat{T}(Q_{\text{new}})$, an agent set $\widehat{A}(Q_{\text{new}})$, and an insight set $\widehat{I}(Q_{\text{new}})$, and we enforce schema-level consistency by meta-path-induced neighborhoods on the heterogeneous schema. We define schema-level neighborhoods induced by the meta-path template $P_{QTQ} : \texttt{Query} \rightarrow \texttt{Tool} \rightarrow \texttt{Query}$ as

$$\mathcal{N}_{QTQ}(Q_{\text{new}}) := \Big\{ q \in \mathcal{Q}_{\text{new}}^S \big| \exists\, t \in \widehat{T}(Q_{\text{new}}), (q, t) \in \mathcal{E}_{QT} \Big\}.$$

For the other two meta-path templates, $P_{QIQ} : \texttt{Query} \rightarrow \texttt{Insight} \rightarrow \texttt{Query}$ and $P_{QAQ} : \texttt{Query} \rightarrow \texttt{Agent} \rightarrow \texttt{Query}$, the corresponding schema-level neighborhoods $\mathcal{N}_{QIQ}(Q_{\text{new}})$ and $\mathcal{N}_{QAQ}(Q_{\text{new}})$ are defined analogously. We refine retrieval by intersecting semantic and behavioral constraints:

$$\tilde{\mathcal{Q}}_{\text{new}} = \mathcal{N}_{QTQ}(Q_{\text{new}}) \cap \mathcal{N}_{QIQ}(Q_{\text{new}}) \cap \mathcal{N}_{QAQ}(Q_{\text{new}}),$$

yielding a compact candidate set for subsequent execution-trace retrieval.

**Memory Subgraph Construction.** For each retrieved query $Q \in \tilde{\mathcal{Q}}_{\text{new}}$, we extract its $h$-hop trace subgraph $\mathcal{G}_Q^{(h)} = (\mathcal{V}_Q^{(h)}, \mathcal{E}_Q^{(h)})$ from the global heterogeneous graph $\mathcal{G}$, where $\mathcal{V}_Q^{(h)}$ denotes nodes reachable from $q$ within $h$ hops under trace-related relations. The trace subgraph focuses on three trace-bearing entity types: tools, agents, and insights, i.e.,

$$\mathcal{T}_Q^{(h)} := \mathcal{V}_Q^{(h)} \cap \mathcal{T}, \quad \mathcal{A}_Q^{(h)} := \mathcal{V}_Q^{(h)} \cap \mathcal{A}, \quad \mathcal{I}_Q^{(h)} := \mathcal{V}_Q^{(h)} \cap \mathcal{I}.$$

Accordingly, each retrieved query $Q$ serves as an anchor node that simultaneously connects to its tool-, agent-, and insight-specific trace neighborhoods via heterogeneous edges, while the within-type dependencies are captured by homogeneous edges among $\mathcal{T}_Q^{(h)}$, $\mathcal{A}_Q^{(h)}$, and $\mathcal{I}_Q^{(h)}$. The memory subgraph for $Q_{\text{new}}$ is obtained by taking the union of trace subgraphs extracted from all retrieved queries:

$$\bar{\mathcal{G}}(Q_{\text{new}}) = \bigcup_{Q \in \tilde{\mathcal{Q}}_{\text{new}}} \mathcal{G}_Q^{(h)}.$$

This memory subgraph summarizes the tool usage, agent collaboration, and reusable insights that are most relevant to $Q_{\text{new}}$, and serves as the structured context for subsequent multi-agent planning and execution.

**Dynamic Task Allocation.** After constructing the heterogeneous memory graph, we summarize its required capability by aggregating the capability requirements of the retrieved queries:

$$C_{\text{new}}^{\mathcal{Q}} := \text{Aggregate}\Big( \big\{ C_i^{\mathcal{Q}} \mid Q_i \in \tilde{\mathcal{Q}}_{\text{new}} \big\} \Big).$$

Then we select the agent whose capability vector best matches $C_{\text{new}}^{\mathcal{Q}}$ under a kernelized compatibility score:

$$a_{i^*} = \arg \max_{a_i \in \mathcal{A}} \mathcal{K}\big( C_{\text{new}}^{\mathcal{Q}}, C_i^{\mathcal{A}} \big).$$

Once the agent $a_{i^*}$ is assigned the task, it processes the memory graph $\bar{\mathcal{G}}(Q_{\text{new}})$ to execute the task:

$$\big( y_{\text{new}}, \tau_{\text{new}} \big) := \text{Execution}\big( a_{i^*}, Q_{\text{new}}, \bar{\mathcal{G}}(Q_{\text{new}}) \big),$$

where $y_{\text{new}}$ denotes the response for the query $Q_{\text{new}}$, and $\tau_{\text{new}}$ records the execution trace (e.g., tool calls, intermediate rationales, and created insights), which is subsequently written back to the dynamic update mechanisms of the heterogeneous graph in Section 5.

We instantiate $\text{Execution}(\cdot)$ with one of the following mutually exclusive strategies: (i) *Direct*: solve $Q_{\text{new}}$ in a single pass conditioned on $\bar{\mathcal{G}}(Q_{\text{new}})$, returning $y_{\text{new}}$; (ii) *Decompose*: initial agent $a_{i^*}$ decomposes $Q_{\text{new}}$ into sub-queries $\{Q_{\text{new}}^{(j)}\}$, solves them, and aggregates their results; (iii) *Collaborative*: the assigned agent forms a temporary team of peers $\mathcal{A}_{\text{col}} \subseteq \mathcal{A}$ to jointly deliberate on $Q_{\text{new}}$, and fuses their candidate solutions into the final output. The execution strategy is chosen by analyzing the mismatch between the assigned agent's capability vector $C_{i^*}^{\mathcal{A}}$ and the task requirement $C_{\text{new}}^{\mathcal{Q}}$: small gaps favor direct execution, whereas larger gaps trigger either decomposition or collaboration depending on the magnitude of $C_{i^*}^{\mathcal{A}}$.

## 5. Dynamic Update Mechanism

In this section, we present the dynamic update mechanism of SCHEMAS, which writes back each completed query and its execution trace to continually refine the heterogeneous memory graph. The key idea is to convert transient interactions (e.g., tool calls, agent collaboration, and emergent insights) into persistent, structured signals that update both node attributes and relation weights.

**Query Graph Update.** After solving the new query $Q_{\text{new}}$, we update the query graph $\mathcal{G}_{\text{query}} = (\mathcal{Q}, \mathcal{E}_{\text{query}})$ by (i) inserting a new query node and (ii) linking it to semantically similar historical queries.

We refine the prior requirement vector $C_{\text{new}}^{\mathcal{Q}}$ using posterior execution signals contained in the trace $\tau_{\text{new}}$ (e.g., confidence):

$$C_{\text{new}}^{\mathcal{Q}} \leftarrow \text{Calibrate}\big( C_{\text{new}}^{\mathcal{Q}}, \tau_{\text{new}} \big).$$

This is well-motivated in well-calibrated MAS, where the reported confidence is predictive of empirical correctness, thus providing a reliable posterior cue for adjusting the estimated capability demand. We then append $q_{\text{new}}$ to the node set:

$$q_{\text{new}} \leftarrow (Q_{\text{new}}, C_{\text{new}}^{\mathcal{Q}}, \Psi_{\text{new}}), \qquad \mathcal{Q} \leftarrow \mathcal{Q} \cup \{q_{\text{new}}\},$$

where $Q_{\text{new}}$ is the task description and $\Psi_{\text{new}}$ records its execution status. Finally, we connect $q_{\text{new}}$ to its top-$k$ most similar historical queries $\mathcal{Q}_{\text{new}}^{\mathcal{S}}$ and update

$$\mathcal{E}_{\text{query}} \leftarrow \mathcal{E}_{\text{query}} \cup \{(q_{\text{new}}, q_i) : q_i \in \mathcal{Q}_{\text{new}}^{\mathcal{S}}\}.$$

**Agent Graph Update.** The agent graph $\mathcal{G}_{\text{agent}} = (\mathcal{A}, \mathcal{E}_{\text{agent}})$ is updated to accommodate the evolving task demands. Since the node set is fixed, we only update edges and agent capability vectors. The agent connections evolve based on collaborative success. We update the weights by:

$$W_{\text{new}}^{\mathcal{A}}(i,j) = \alpha \, W^{\mathcal{A}}(i,j) + (1 - \alpha) \, \text{Coop}(a_i, a_j; \tau_{\text{new}}),$$

where $\alpha \in [0, 1]$ is a decay factor, $\text{Coop}(a_i, a_j; \tau_{\text{new}})$ extracts a collaboration signal from the execution trace, and assigns a larger value when $a_i$ and $a_j$ collaborate more effectively on the current query $Q_{\text{new}}$. The edge set $\mathcal{E}_{\text{new}}$ for the updated agent graph is determined by:

$$\mathcal{E}_{\text{new}}^{\mathcal{A}} = \left\{ (a_i, a_j) \mid W_{\text{new}}^{\mathcal{A}}(i,j) > \varepsilon \right\},$$

where $W_{\text{new}}^{\mathcal{A}}(i,j)$ represents the interaction weight between agents $a_i$ and $a_j$, and $\varepsilon$ is a threshold that prunes unproductive connections.

Following task execution, the capabilities of the participating agents are updated based on their task performance. The updated capability vector for agent $a_i \in \mathcal{A}$ is given by:

$$C_i^{\mathcal{A}} \leftarrow \beta \cdot C_i^{\mathcal{A}} + (1 - \beta) \cdot \Delta C_i^{\mathcal{A}}(y_{\text{new}}, \tau_{\text{new}}),$$

where $\beta \in [0, 1]$ is a decay factor and $\Delta C_i^{\mathcal{A}}(y_{\text{new}}, \tau_{\text{new}})$ denotes the capability gain demonstrated by $a_i$ on $Q_{\text{new}}$, computed from the output quality and the executed operation types recorded respectively in $(y_{\text{new}}, \tau_{\text{new}})$. This rule fuses the agent's prior capability profile with newly observed task evidence, progressively refining $C_i$ for more accurate task allocation.

**Tool Graph Update.** At initialization, we predefine a tool inventory $\mathcal{T}$ according to the available APIs and tool interfaces in the system. We initialize the edge set $\mathcal{E}_{\text{tool}}$ by connecting each tool to its top-$k$ most semantically similar tools, where semantic similarity is computed from static tool descriptors $\xi_i$. We then construct a global tool graph $G_{\text{tool}} = (\mathcal{T}, \mathcal{E}_{\text{tool}})$ and associate a weighted matrix $W^{\mathcal{T}} \in \mathbb{R}^{|\mathcal{T}| \times |\mathcal{T}|}$, where $W^{\mathcal{T}}(i,j)$ quantifies the strength of composing tool $t_i$ followed by $t_j$.

We update the tool graph in two aspects: node attribute update and edge weight update. For each tool $t_i$ used in $\tau_{\text{new}}$, we compute a task-induced statistic increment $\Delta \zeta_i$ from $\Gamma_{m+1}$ and apply an update:

$$\zeta_i \leftarrow \lambda \zeta_i + (1 - \lambda) \, \Delta \zeta_i(y_{\text{new}}, \tau_{\text{new}}),$$

where $\lambda \in [0, 1]$ is a decay factor, $\Delta \zeta_i(y_{\text{new}}, \tau_{\text{new}})$ summarizes task-induced evidence from the output and trace to update dynamic tool statistics (e.g., usage count, success rate, and cost).

We update the weighted adjacency $W^{\mathcal{T}}$ based on adjacent tool transitions in the trace. Specifically, for each consecutive pair $(t_i, t_j)$ appearing in $\tau_{\text{new}}$, we accumulate a transition signal

$$\Delta W^{\mathcal{T}}(i,j) = \text{Trans}(t_i \rightarrow t_j; \, y_{\text{new}}, \tau_{\text{new}}),$$

and apply an exponential moving average update:

$$W^{\mathcal{T}}(i,j) \leftarrow \rho \, W^{\mathcal{T}}(i,j) + (1 - \rho) \, \Delta W^{\mathcal{T}}(i,j),$$

where $\rho \in [0, 1]$ is a decay factor. Here $\text{Trans}(\cdot)$ extracts the strength of composing $t_i$ followed by $t_j$ from $\tau_{\text{new}}$ (e.g., co-occurrence frequency, successful completion, and execution cost), yielding larger updates for more reliable and effective tool compositions. For pairs not observed in $\tau_{\text{new}}$, we keep their weights unchanged.

**Remark.** The tool graph is maintained in an incremental manner: whenever an unseen tool appears in the execution trace, we insert it as a new node with metadata-based initialization and connect it to existing tools via semantic similarity and/or trace co-occurrence, after which its node statistics and edge weights are updated online.

**Insight Graph Update.** We update the insight graph after each completed query to continuously accumulate reusable strategies. Recall that $\mathcal{G}_{\text{insight}} = (\mathcal{I}, \mathcal{E}_{\text{insight}})$. We distill a set of candidate insights after executing $Q_{\text{text}}$:

$$\widehat{\mathcal{I}} = \text{Distill}(y_{\text{new}}, \tau_{\text{new}}),$$

where each $\widehat{\iota} \in \widehat{\mathcal{I}}$ is associated with candidate strategy content $\widehat{\kappa}$. For each candidate $\widehat{\iota}$, we match it to existing insights via semantic similarity:

$$\iota_{i^*} = \arg\max_{\iota_i \in \mathcal{I}} \mathcal{K}(z(\widehat{\kappa}), z(\kappa_i))$$

If the best score exceeds a threshold $\eta$, we merge $\widehat{\iota}$ into $\iota_{i^*}$ by updating its support set and refining the strategy content:

$$\Omega_{i^*} \leftarrow \Omega_{i^*} \cup \{q_{\text{new}}\}, \qquad \kappa_{k^*} \leftarrow \text{Merge}(\kappa_{k^*}, \widehat{\kappa}).$$

Otherwise, we insert a new insight node $\iota_{\text{new}}$ with

$$\iota_{\text{new}} \leftarrow (\widehat{\kappa}, q_{\text{new}}), \qquad \mathcal{I} \leftarrow \mathcal{I} \cup \{\iota_{\text{new}}\}.$$

We add inter-insight edges according to the observed strategy organization in $\tau_{\text{new}}$. Specifically, if the execution trace indicates a semantic association between a candidate insight $\widehat{\iota}_i$ and another candidate insight $\widehat{\iota}_j$, we add a directed edge:

$$\mathcal{E}_{\text{insight}} \leftarrow \mathcal{E}_{\text{insight}} \cup \{(\widehat{\iota}_i, \widehat{\iota}_j, q_{\text{new}})\},$$

where $\widehat{\iota}_i, \widehat{\iota}_j$ denote the corresponding merged/inserted nodes in $\mathcal{I}$.

**Global Heterogeneous Graph Update.** To make retrieval traceable from a matched query to its execution experience, we augment the heterogeneous graph by linking the new query node $q_{\text{new}}$ to the entities involved in its execution trace. We then update the heterogeneous edge sets as

$$\mathcal{E} \leftarrow \mathcal{E} \cup \{(q_{\text{new}}, a_{i*}), (q_{\text{new}}, t), (q_{\text{new}}, \iota) : t \in \tau_{\text{new}}, \iota \in \widehat{\mathcal{I}}\}.$$

These heterogeneous graph associations connect the query graph with the content layer (agent/tool/insight graphs), enabling the system to retrieve not only similar past queries but also their corresponding execution traces for reuse.

# 6. Experiments

In this section, we conduct extensive experiments to answer the following research questions (RQs): (1) Can SCHEMAS deliver strong and consistent performance on challenging tool use benchmarks? (2) How does SCHEMAS's structured memory compare with existing memory baselines? (3) Does SCHEMAS incur excessive resource overhead to achieve its performance gains? (4) How sensitive is SCHEMAS to its key components?

## 6.1. Experimental Setup

**Datasets.** To comprehensively evaluate SCHEMAS, we conduct experiments along two axes: (1) **general tool use**, which measures tool selection and invocation efficiency, and (2) **interactive tasks**, which stress experience reuse and long-horizon decision-making supported by our structured memory. For tool use, following (Li et al., 2025b), we consider four representative benchmarks: ToolBench (Qin et al., 2023), API-Bank (Li et al., 2023), ToolHop (Ye et al., 2025), and TMDB (Song et al., 2023). For memory evaluation, following (Zhang et al., 2025a), we adopt four widely used benchmarks: ALFWorld (Shridhar et al., 2020), SciWorld (Wang et al., 2022) PDDL (Chang et al., 2024) and HotpotQA (Yang et al., 2018). Detailed descriptions are provided in Appendix A.

**Baselines.** Our baselines fall into two categories: (1) tool use agents, including single-agent baselines ReAct (Yao et al., 2022), CodeAct (Wang et al., 2024a), Plan-and-Solve (Wang et al., 2023c), and DeepAgent-Base (Li et al., 2025b), as well as multi-agent baselines ChatDev (Qian

et al., 2023) and MetaGPT (Hong et al., 2023); and (2) memory-augmented agents, including Voyager (Wang et al., 2023b), MemoryBank (Zhong et al., 2024), Generative Agents (Park et al., 2023), ChatDev (Qian et al., 2023), (Hong et al., 2023) and G-Memory (Zhang et al., 2025a).

**Evaluation Metrics.** For **tool use** tasks, we follow DeepAgent (Li et al., 2025b) and report **Pass@1** (**Success**) and **Path** (average *relevant-API coverage*). For **interactive tasks**, we follow G-Memory (Zhang et al., 2025a). We use exact-match accuracy for HotpotQA (Yang et al., 2018), report the progress rate for SciWorld (Wang et al., 2022) and PDDL (Chang et al., 2024), and use the success rate for ALFWorld (Shridhar et al., 2020).

**MAS and LLMs Backbones.** For a fair comparison in the multi-agent setting, we adopt MacNet (Qian et al., 2024) as the underlying multi-agent framework, a representative decentralized MAS architecture. We configure each multi-agent baseline with five agents. To instantiate these MAS baselines, we use four open-source LLM backbones, `Qwen-2.5-7B`, `Qwen-2.5-14B`, `Qwen-2.5-32B`, and `QwQ-32B`, as well as one proprietary model, `GPT-4o-mini`. For single-agent baselines, we use `Qwen-2.5-32B` as the backbone for all methods.

## 6.2. Experimental Results

▷**Results of the Tool Use Tasks (RQ1).** Table 1 presents results on general tool use, leading to three key observations. ❶ **Consistent gains across tool-use benchmarks.** SCHEMAS achieves the best Pass@1 on all four benchmarks under both single-agent and multi-agent settings. For instance, it surpasses the strongest single-agent baseline by 16.25% on TMDB and outperforms MetaGPT by 12.39% on API-Bank in the multi-agent setting, demonstrating robust improvements across diverse tool-use scenarios. ❷ **Higher tool-path compliance**. SCHEMAS attains the highest Path scores on every benchmark (e.g., 71.43% on API-Bank and 76.39% on TMDB), indicating stronger coverage of ground-truth relevant APIs during execution and more faithful alignment with the intended tool-use pathways. ❸ **Tool use demands specialized memory design**. Our results indicate that structured memory can substantially enhance tool-use performance in MAS. Specifically, SCHEMAS retrieves a query-conditioned subgraph that links semantically similar historical queries to successful tool trajectories. This retrieval facilitates the reuse of tool-selection, mitigates redundant trial-and-error, and is accompanied by higher Pass@1 and Path scores.

▷**Comparison with memory baselines (RQ2).** Table 2 presents results on four interactive benchmarks, leading to several key observations. ❶ **SCHEMAS consistently outperforms memory-augmented baselines.** SCHEMAS achieves the best performance on all benchmarks, improving

*Table 1.* Performance (%) comparison with single- and multi-agent baselines on four general tool-use benchmarks, reporting Pass@1 and Path. The best results are in bold.

| Setting | Methods | ToolBench | | API-Bank | | ToolHop | | TMDB | |
|---|---|---|---|---|---|---|---|---|---|
| | | Success | Path | Success | Path | Correct | Path | Success | Path |
| Single Agent | ReAct | 55.42 | 21.11 | 15.63 | 41.47 | 13.86 | 18.21 | 11.58 | 33.92 |
| | CodeAct | 50.37 | 19.62 | 22.41 | 49.05 | 12.18 | 17.92 | 19.74 | 46.29 |
| | Plan-and-Solve | 54.68 | 19.89 | 18.57 | 43.31 | 12.63 | 15.78 | 15.52 | 41.06 |
| | DeepAgent-Base | 60.71 | 36.24 | 21.39 | 62.44 | 37.85 | 40.97 | 51.46 | 72.38 |
| Multi-Agent | ChatDev | 60.33 | 37.59 | 18.72 | 63.41 | 40.66 | 42.38 | 60.58 | 74.07 |
| | MetaGPT | 63.76 | 38.44 | 22.35 | 67.62 | 40.67 | 46.71 | 61.39 | 74.55 |
| | **SCHEMAS** | **69.58** | **42.31** | **34.74** | **71.43** | **47.62** | **48.47** | **67.71** | **76.39** |

*Table 2.* Performance (%) comparison with memory-augmented baselines and ablations on four benchmarks. The best results are in bold, and "w/o all" removes the agent, insight, and tool graphs.

| Method | ALFWorld | SciWorld | PDDL | HotpotQA |
|---|---|---|---|---|
| Voyager | 46.58 | 29.47 | 11.92 | 25.06 |
| MemoryBank | 52.88 | 27.29 | 12.79 | 24.51 |
| Generative | 47.89 | 31.62 | 13.49 | 24.93 |
| MetaGPT-M | 53.41 | 29.21 | 17.13 | 24.96 |
| ChatDev-M | 45.36 | 25.88 | 10.74 | 16.89 |
| G-Memory | 55.07 | 31.68 | 18.02 | 26.97 |
| SCHEMAS w/o Insight | 59.33 | 35.12 | 20.11 | 27.39 |
| SCHEMAS w/o Agent | 58.72 | 34.43 | 17.03 | 26.01 |
| SCHEMAS w/o Tool | 56.77 | 32.47 | 18.03 | 28.01 |
| SCHEMAS w/o All | 48.51 | 28.95 | 11.04 | 24.49 |
| **SCHEMAS** | **60.34** | **36.78** | **21.89** | **30.12** |

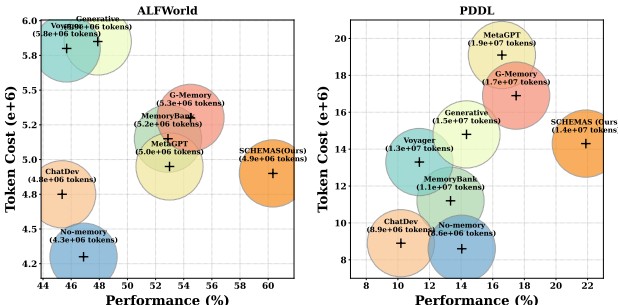

*Figure 3.* Cost analysis of SCHEMAS. We showcase the performance versus the overall system token cost.

*Table 3.* Ablation comparison (%) with **meta-path** (MP) and **dynamic task allocation** (DTA) components.

| MAS | MP | DTA | ALFWorld | SciWorld |
|---|---|---|---|---|
| SCHEMAS | ✓ | ○ | 54.94 | 32.27 |
| | ○ | ✓ | 57.21 | 33.75 |
| | ✓ | ✓ | 60.34 | 36.78 |

over the strongest prior baseline (G-Memory) by 5.27% on ALFWorld, 5.1% on SciWorld, 3.87% on PDDL, and 3.15% on HotpotQA. ❷ **Structured memory and capability-aware routing benefit MAS execution.** Beyond storing past interactions, SCHEMAS organizes experience into a structured, query-conditioned memory graph and matches tasks to agents based on capability demands, enabling targeted retrieval of relevant trajectories and more appropriate agent choices. This reduces irrelevant exploration in interactions, yielding consistent gains.

▷**Cost Analysis (RQ3).** To evaluate the efficiency of SCHEMAS in terms of token consumption, we visualize the performance-token cost trade-off across benchmarks, as shown in Figure 3. SCHEMAS achieves strong performance without excessive token consumption, demonstrating its token efficiency.

▷**Ablation Study (RQ4).** Table 2 and Table 3 present our ablation results, leading to three key observations. ❶ **Removing any memory subgraph degrades performance.** These results indicate that these subgraphs provide complementary signals for retrieval and execution in interactive settings. ❷ **Meta-path reasoning is crucial for effective retrieval.** Disabling meta-paths (w/o MP) leads to clear performance degradations on both ALFWorld and SciWorld, showing that structurally constrained retrieval is important for selecting relevant experience beyond purely semantic

matching. ❸ **Dynamic task allocation further improves multi-agent coordination.** Removing DTA (w/o DTA) also reduces performance, while enabling both MP and DTA achieves the best results, suggesting that capability-aware, dynamic routing helps assign tasks to suitable agents and reduces redundant interactions.

# 7. Conclusion

In this paper, we present SCHEMAS, a scalable heterogeneous framework for MAS that unifies query, agent, tool, and distilled insight within a single graph. Furthermore, we propose a structured memory mechanism that performs query-conditioned subgraph construction and capability-aware dynamic task allocation, enabling more effective experience reuse in interactions. Extensive experiments across eight benchmarks demonstrate that SCHEMAS consistently outperforms strong baselines with favorable efficiency. We hope that SCHEMAS will inspire future research on structured, self-evolving memory for reliable and scalable multi-agent intelligence.

## 8. Impact Statement

This paper studies structured memory for multi-agent systems (MAS) and proposes SCHEMAS, a scalable heterogeneous graph framework that supports query-conditioned retrieval, capability-aware routing, and continual trace updates; by improving experience reuse and tool-path compliance, SCHEMAS can reduce redundant trial-and-error, lower resource consumption, and increase reliability in long-horizon tool-use applications. However, the memory graph may store sensitive information contained in interaction traces, raising privacy concerns if deployed with real user data, and over-reliance on historical patterns may harm robustness under distribution shifts or adversarial tool outputs. We therefore urge responsible deployment of SCHEMAS with appropriate safeguards, including continual validation, adversarial robustness checks, and alignment with human values, to ensure that structured memory improves MAS reliability while limiting unintended harms.

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

# A. Experimental Details

## A.1. Benchmarks

We conduct extensive experiments on a wide range of benchmarks, including tool-use tasks and interactive tasks.

**General Tool-Use.** These benchmarks cover tool sets ranging from dozens to more than ten thousand, providing a challenging testbed for evaluating how well different methods scale to large and diverse tool inventories.

- **ToolBench** (Qin et al., 2023): A large-scale collection of real-world REST APIs (over 16,000) spanning 49 categories. Its test split contains 100 cases that probe both single-tool usage and more involved multi-tool compositions.

- **API-Bank** (Li et al., 2023): A comprehensive suite for tool-augmented LLMs with an executable evaluation pipeline. It includes 73 API tools and a large training corpus (over 2,200 dialogues) covering 2,211 APIs from 1,008 domains, targeting planning, retrieval, and API invocation abilities.

- **TMDB** (Song et al., 2023): A RestBench sub-scenario centered on the TMDB movie database, consisting of 100 questions that call 54 local tools and typically require 2.3 sequential API calls.

- **ToolHop** (Ye et al., 2025): A multi-hop reasoning dataset with 995 challenging questions, backed by 3,912 locally executable tools and requiring 3-7 successive tool calls per instance.

**Interactive tasks.** To evaluate whether agents can reuse past experience and sustain long-horizon reasoning, we adopt four widely used benchmarks that require persistent state tracking, multi-step planning, and evidence accumulation.

- **ALFWorld** (Shridhar et al., 2020) is a text-based embodied household environment. Agents must navigate rooms and manipulate objects via natural-language actions, which tests long-horizon planning, spatial grounding, and error recovery.

- **ScienceWorld** (Wang et al., 2022) is a text-based embodied environment for interactive science tasks. Agents explore rooms, operate equipment, and perform experiments, stressing procedural reasoning, hypothesis-driven exploration, and multi-step tool interaction.

- **PDDL** (Chang et al., 2024) is a game dataset from AgentBoard (Chang et al., 2024) comprising strategic games specified in PDDL. Agents must reason over symbolic task descriptions and produce action sequences that satisfy constraints, making it suitable for evaluating structured planning and compositional generalization.

- **HotpotQA** (Yang et al., 2018) is a multi-hop question answering benchmark with supervision on supporting facts. It evaluates the ability to retrieve, synthesize, and justify answers from multiple documents, particularly when coupled with web-search tools for evidence-based reasoning.

## A.2. Baselines

- **ReAct** (Yao et al., 2022): A general agent paradigm that tightly couples reasoning with tool interaction. It elicits an interleaved trajectory of *reasoning* (natural-language thoughts), *actions* (tool calls), and *observations* (tool outputs), enabling the agent to iteratively refine its plan based on intermediate feedback.

- **CodeAct** (Wang et al., 2024a): A code-centric agent framework where decisions are expressed as executable Python programs and run in an interpreter. Representing actions in code provides stronger compositionality and controllability, allowing the agent to orchestrate tool usage, invoke APIs, and perform multi-step computations through program execution.

- **Plan-and-Solve** (Wang et al., 2023c): A plan-first baseline that separates deliberation from execution. The agent first drafts a structured, step-by-step solution plan without external tool calls, and then follows the plan to carry out the required computations or actions, offering a simple but effective approach for tackling multi-step tasks.

- **DeepAgent-Base** (Li et al., 2025b): An autonomous deep reasoning agent that integrates tool discovery and tool execution within a single, coherent reasoning process. It supports on-demand tool search over scalable toolsets and can optionally compress long interaction histories into structured memories via a memory folding mechanism, mitigating context-length explosion and error accumulation in long-horizon tool use. We use the **Base** version (without end-to-end ToolPO training) as a strong agentic baseline.

- **Voyager** (Wang et al., 2023b): It is an embodied agent that incrementally builds experience by interacting with an environment and producing new artifacts, where memory plays a central role in continual improvement. As the original design targets a single-agent setting, we extend it to multi-agent execution by enabling per-agent history retrieval conditioned on each agent's observable dialogue context.We follow the same adaptation principle for other single-agent memory baselines, ensuring fair comparisons under the same MAS interface.

- **MemoryBank** (Zhong et al., 2024): A memory system that emulates human-like retention and forgetting through selective storage. It updates memories using an Ebbinghaus-style temporal decay mechanism, where entries are strengthened or pruned based on both recency and an estimated importance score, thereby controlling memory growth over time.

- **Generative Agents (Generative)** (Park et al., 2023): A baseline that maintains two complementary memory types: (i) raw observational traces and (ii)reflective memory. he reflective component stores higher-level summaries and abstracted thoughts produced via reflection, aiming to capture reusable patterns beyond verbatim observations.

- **MetaGPT-M** (Hong et al., 2023): We follow the memory setting used in G-Memory (Zhang et al., 2025a) and focus solely on *inside-trial* memory, i.e., information stored internally during the resolution of a single task by multiple agents.

- **ChatDev-M** (Qian et al., 2023): We follow the memory setting used in G-Memory (Zhang et al., 2025a), which incorporates both inside-trial and cross-trial memory. The inside-trial memory is passed from the central (or initiating) agent at the beginning of each round to provide guidance based on prior interactions. The cross-trial memory is relatively simple, storing past solutions to previous queries for future retrieval; however, in our tasks, it does not effectively manage information-rich inter-agent collaboration.

- **G-Memory** (Zhang et al., 2025a): A hierarchical memory framework for multi-agent systems that organizes experience into multiple graph layers (e.g., insight, query, and interaction graphs) and retrieves reusable collaboration patterns via cross-layer traversal. It enables coarse-to-fine memory retrieval and supports continual updates by writing back new trajectories, serving as a strong structured-memory baseline.

### A.3. Parameter Settings.

Unless otherwise specified, we use the following default settings across all experiments. For structured retrieval, we embed each query with an off-the-shelf sentence encoder $f_\theta$ and compute kernelized similarity scores $K(\cdot, \cdot)$. We instantiate $f_\theta$ with MiniLM (Wang et al., 2020) and use the cosine kernel by default. For coarse retrieval, we select the top-$k$ similar historical queries with $k = 7$. For fine-grained retrieval, we construct meta-path constrained neighborhoods using $P_{QTQ}$, $P_{QIQ}$, and $P_{QAQ}$, and extract $h$-hop trace subgraphs with $h = 2$ for each retrieved query before taking their union as the query-conditioned memory subgraph. For dynamic updates, we prune edges with weights below $\varepsilon = 0.1$. For insight merging, we match candidate insights to existing ones by cosine similarity and merge them when the similarity exceeds $\eta = 0.8$; otherwise we insert a new insight node.

## B. Additional Graph Structures in MAS

In this section, we provide an illustrative example (Figure 4) to show that, beyond our core query-agent-tool-insight graph, MAS can also benefit from additional graph abstractions that structure heterogeneous entities and relations for downstream reasoning and orchestration. Looking ahead, our unified heterogeneous graph is compatible with incorporating such auxiliary graphs as additional node/edge types (e.g., environment states and factual knowledge), enabling richer memory organization and more efficient coordination; we leave a systematic integration and evaluation of these extended structures to future work.

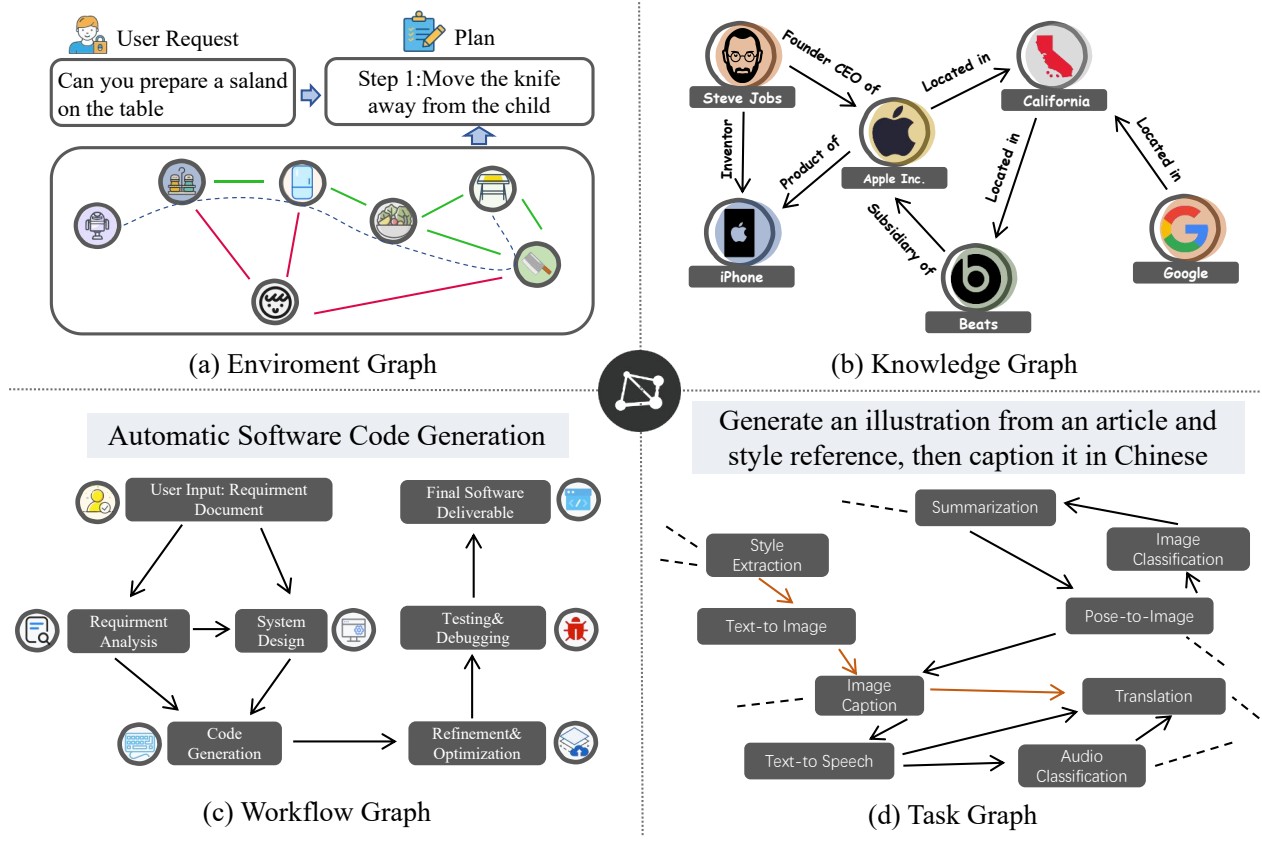

*Figure 4.* Exemplars of additional graphs in MAS: environment, knowledge, workflow, and task graphs.

## C. Prompts for Our SCHEMAS

**Prompt for API Call**

```
You are an API orchestration agent that generates EXACTLY ONE tool call based on
    conversation context for the next step from APIs.

YOUR TASK is Generate a SINGLE valid API call that:
  - Extracts ALL required parameters DIRECTLY from conversation history.
  - Preserves original wording EXACTLY (no paraphrasing/modification).
  - Uses token from [Latest API Result] when required.
  - Matches parameter names CASE-SENSITIVELY to API schema.
  - Formats input parameters and fields verbatim as provided.

STRICT OUTPUT RULES (NON-NEGOTIABLE):
  - Output MUST be exactly ONE line starting with "[" and ending with "]".
  - Format: [ApiName(param1='verbatim_value1', param2='verbatim_value2')].
  - NEVER include: explanations/comments/analysis/multiple calls/JSON.
  - NEVER modify user-provided strings.
  - NEVER infer missing parameters-omit call if requirements unmet.

MISSING-INPUT RULE FOR TOOL USE:
  - If a tool is needed but you do not have valid concrete values for all required
      inputs of the intended tool, you must first rely on other available tools in
      the Tool Library to obtain those missing input values.

[Tools Descriptions]
{tools_descriptions}
```

```
OUTPUT FORMAT:
[ApiName(key1='value1', key2='value2', ...)]

You will be given:
 1) [API Descriptions]
 2) [Top-K Related Tasks]
 3) [Retrieved Task Subgraphs]
 4) [Recommended Tools]

[API Descriptions]
{api_descriptions}

[Top-K Related Tasks]
{top_k_tasks}

[Retrieved Task Subgraphs]
{retrieved_subgraphs}

[Recommended Tools]
{recommended_tools}

[Latest API Result]
{api_result}
```

**Prompt for Simply Response**

```
You are an AI assistant that generates natural language responses follow the latest
    API calls.

YOUR TASK is Generate exactly ONE conversational response that:
  - Acknowledges the API result naturally (without exposing technical details) from [
    Latest API Result].
  - Provides helpful next steps or confirmation.
  - Uses friendly yet professional tone.
  - Stays concise (under 2 sentences).
  - You need to carefully reflect on your answers based on your task and assign an
    objective confidence score to your results. The confidence score should be a
    two-digit floating-point number within the interval [0,1].

STRICT RULES:
  - Output MUST contain ONLY that single line.
  - NEVER include: JSON/tool syntax/parameters/status codes/analysis/reasoning.
  - NEVER reference internal context sections (e.g., "based on the API description
    ...").
  - Respond as a human assistant would in a real conversation.

OUTPUT FORMAT (STRICT):
AI: <natural language response>
Confidence Score: <two-digit floating-point number>

You will be given:
 1) [API Descriptions]
 2) [Top-K Related Tasks]
 3) [Retrieved Task Subgraphs]
 4) [Recommended Tools]

[API Descriptions]
{api_descriptions}

[Top-K Related Tasks]
```

```
880    {top_k_tasks}
881
882    [Retrieved Task Subgraphs]
883    {retrieved_subgraphs}
884
885    [Recommended Tools]
886    {recommended_tools}
887
888    [Latest API Result]
       {api_result}
889
```

**Prompt for Decision Making**

```
892    You are a task-oriented agent. Your job is to decide the next step based on the
893        current task and context.
894
895    [INPUT]
896     1) Task Description:
        {task_description}
897
898     2) Task Context:
899     {task_context}
900
901     3) Tool Description (the complete list of available tools and their descriptions,
902         including required inputs):
        {tool_description}
903
904    [DECISION RULE]
905     - Output exactly ONE label.
906     - Output "ai_response" if the task can be completed using only the information
907         already present in the Task Context.
        - Output "api_call" if essential information is missing and a tool must be used.
908
909    [OUTPUT FORMAT](Do not output anything else)
910    Output exactly one token: "api_call" or "ai_response".
```

**Prompt for Insights Extraction**

```
913    You are an insight distiller. You will extract reusable, task-agnostic execution
914        insights from a completed task trajectory.
915
916    You are given (1) a task description and (2) a full trajectory (including thinking
917        notes, tool calls, intermediate results, and step count), produce a set of
918        independent, reusable insights that can help solve similar tasks in the future.
919
920    Insights must be:
921     - Independent: each insight can be applied without relying on other insights.
922     - Reusable: phrased as general strategies, patterns, or checklists, not tied to
923         this single instance.
924     - Actionable: includes triggers, steps, and tool usage guidance.
        - Non-redundant: no two insights should say the same thing.
925
926    [INPUTS]
927    [Task Description]
        {task_description}
928
929    [Full Trajectory]:
930    {trajectory}
931
932    [INSIGHT EXTRACTION RULES]
933     1) Create 2-8 insights (choose the number that best matches the trajectory
           complexity).
934
```

```
  2) Do NOT force one insight per step; instead group steps into coherent reusable
     patterns.
  3) Each insight MUST reference evidence from the trajectory using step indices/IDs.
  4) Do NOT include user-private identifiers, credentials, API keys, or any sensitive
     data.
  5) Do NOT expose chain-of-thought or internal policies. If trajectory contains
     thoughts, abstract them into safe heuristics.
  6) Each insight must include:
   - Trigger condition (when to use it).
   - Procedure (how to do it).
   - Tool strategy (which tools, what to extract).
   - Pitfalls (common failure modes).
   - A reusable template (a short fill-in-the-blank recipe or pseudo-steps).

[OUTPUT FORMAT]
Output MUST be a single JSON object and nothing else.

Schema:
{
  "task_summary": "<1-2 sentences summarizing what was accomplished>",
  "insights": [
    {
      "id": "I1",
      "title": "<concise, reusable title>",
      "trigger": "<when this insight applies>",
      "procedure": [
       "<step-like action 1>",
       "<step-like action 2>",
       "<...>"
      ],
      "tool_strategy": [
        {
          "tool_name": "<tool used>",
          "what_to_get": "<the exact type of information or artifact to retrieve>",
          "how_to_use_output": "<how the tool output should be consumed next>"
        }
      ],
      "pitfalls": [
       "<pitfall 1>",
       "<pitfall 2>"
      ],
      "reusable_template": "<a short generic template/checklist with placeholders>"
    }
   ...
}

Now produce the JSON.
```

### Prompt for Intent Sketch Generation

```
You are an Intent Sketch Generator for a memory-augmented multi-agent system.
Your goal is to produce a coarse solution sketch for the NEW query Q_new, and then
   extract the minimal predicted sets of Tools, Agents, and reusable Insights
   implied by that sketch.
These predicted sets will be used downstream for schema-level consistency filtering
   via meta-path-induced neighborhoods.

You must follow the provided catalogs strictly:
 - Do NOT invent tools/agents/insights or IDs.
 - Be conservative and prefer minimal sufficient sets.
 - If uncertain, include at most the top 3 candidates and provide calibrated
     confidence.
```

```
[INPUTS]
[New Query]
{Q_new}

[AGENT_CATALOG]
(each item: {agent_id, name, short_profile, capability_tags})
{agent_catalog}

[TOOL_CATALOG]
(each item: {tool_id, name, description, io_signature, constraints})
{tool_catalog}

[INSIGHT_CATALOG]
(each item: {insight_id, title, content, applicability_tags})
{insight_catalog}

[TWO-PHASE TASK]
Phase A – Draft a coarse solution sketch (plan first, sets second):
  A1) Interpret Q_new: identify task type, deliverable format, constraints, and
      success criteria.
  A2) Produce a step-by-step solution sketch with 4-10 steps.
     - Each step must specify:
       (i) what to do,
       (ii) why it is needed,
       (iii) what information/artifact it produces.
     - If a step requires an external operation (e.g., browsing, calculation, code
         execution), mark it as "requires_tool": true.
  A3) Assign responsibility: for each step, select the most suitable agent_id from
      AGENT_CATALOG (do not invent roles).

Phase B – Extract predicted entity sets from the sketch:
  B1) Tool set extraction: include a tool iff at least one step has "requires_tool":
      true AND the tool is necessary for the sketch.
  B2) Agent set extraction: include an agent iff it is assigned at least one step AND
       is necessary (merge redundant agents).
  B3) Insight set extraction: include an insight iff it is explicitly applicable to
      at least one step (e.g., reusable strategy/pattern/template).
  B4) Calibrate confidence: assign confidence in [0,1] for each selected entity,
      reflecting necessity + relevance (not just similarity).

[STRICT SELF-CHECK]
 1) Every selected ID must exist in the provided catalogs.
 2) Prefer minimal sets: keep each of T_hat, A_hat, I_hat to at most 3 items unless
    Q_new clearly needs more.
 3) Do not include entities just in case; each must be justified by specific step
     indices in the solution sketch.
 4) Output MUST be valid JSON and nothing else (no prose, no markdown).

[OUTPUT FORMAT]
Output MUST be a single JSON object and nothing else.

Schema:
{
  "query_understanding": {
   "task_type": "<string>",
   "constraints": ["<string>", "..."],
   "success_criteria": ["<string>", "..."]
  },
  "solution_sketch": [
    {
     "step_id": 1,
     "action": "<what to do>",
```

```
1045        "rationale": "<why needed>",
1046        "artifact": "<what it produces>",
1047        "requires_tool": true/false,
1048        "assigned_agent_id": "<agent_id from AGENT_CATALOG>",
1049        "candidate_tool_ids": ["<tool_id>", "..."],
1050        "candidate_insight_ids": ["<insight_id>", "..."]
1051      }
1052    ],
1053    "extracted_sets": {
1054      "T_hat": [
1055        {
1056          "tool_id": "<tool_id>",
1057          "confidence": 0.xx,
1058          "supported_by_steps": [1, 3],
1059          "rationale": "<one concrete sentence tied to the sketch>"
1060        }
1061      ],
1062      "A_hat": [
1063        {
1064          "agent_id": "<agent_id>",
1065          "confidence": 0.xx,
1066          "supported_by_steps": [1, 2],
1067          "rationale": "<one concrete sentence tied to the sketch>"
1068        }
1069      ],
1070      "I_hat": [
1071        {
1072          "insight_id": "<insight_id>",
1073          "confidence": 0.xx,
1074          "supported_by_steps": [2],
1075          "rationale": "<one concrete sentence tied to the sketch>"
1076        }
1077      ]
1078    },
1079    "notes": {
1080      "assumptions": ["<at most 2>", "..."],
1081      "uncertainties": ["<at most 2>", "..."]
1082    }
1083  }

     [CONFIDENCE RUBRIC]
      - 0.85-1.00: indispensable for the sketch to work
      - 0.60-0.85: highly likely but may have substitutes
      - 0.35-0.60: plausible optional fallback (avoid including many)

     Now produce the JSON.
```

