# OpenReview forum: "SCHEMAS: Leveraging Scalable Heterogeneous Graph for Query-Guided Reasoning in Multi-Agent Systems"
_ICML.cc/2026/Conference — Submitted to ICML 2026_

### Official Review · Reviewer_tKPB · 2026-02-26

**Soundness:** 3
**Presentation:** 3
**Significance:** 3
**Originality:** 2
**Overall Recommendation:** 4
**Confidence:** 4

**Summary:**

This paper introduces SCHEMAS, a framework that organizes multi-agent systems using a scalable heterogeneous graph to unify four entity types: queries, agents, tools, and distilled insights. By explicitly modeling typed interactions, it addresses the fragmentation of implicit dependencies in current systems. The framework employs a query-driven structured memory mechanism for coarse-to-fine subgraph retrieval and matches query requirements to agent capability profiles for dynamic task allocation. After execution, a dynamic update mechanism writes traces back to the graph, enabling continual self-evolution. Extensive experiments across eight benchmarks show that SCHEMAS consistently outperforms baselines in both accuracy and token efficiency.

**Compliance With Llm Reviewing Policy:**

Affirmed.

**Final Justification:**

This is a fairly solid work. I think it essentially meets the acceptance bar for the conference, and the rebuttal addressed the vast majority of my concerns.

**Key Questions For Authors:**

See weaknesses above.

**Limitations:**

Yes, the authors have adequately discussed the limitations and potential negative societal impacts in Section 8.

**Strengths And Weaknesses:**

## Strengths

- The paper does a great job of unifying queries, agents, tools, and insights into a single, extensible heterogeneous graph schema. This is a significant step up from prior works that focus on isolated components like tool-only or agent-only graphs.
- The "coarse-to-fine" retrieval strategy, combining semantic kernelized similarity with meta-path consistency, offers a very sensible and structured way to handle experience reuse in Multi-Agent Systems.
- The framework is evaluated across eight different benchmarks covering both general tool-use (e.g., ToolBench, API-Bank) and interactive tasks (e.g., ALFWorld, HotpotQA), demonstrating its robustness across diverse scenarios.
- SCHEMAS achieves state-of-the-art performance while maintaining lower token consumption compared to strong baselines like MetaGPT and G-Memory, highlighting its practical utility.
- The trace-driven update mechanism enables the system to "learn" from every execution by updating node attributes and relation weights, turning transient interaction data into long-term structured experience.

## Weaknesses

### Methodological Concerns

- **Complexity and Scalability Analysis**: I’m a bit concerned about the overhead of maintaining such a large heterogeneous graph, especially in scenarios like ToolBench with over 16,000 APIs. The paper lacks a detailed theoretical or empirical analysis of memory footprint, computational latency, and the algorithmic complexity of the frequent graph update process.
- **Under-specified Capability Vectors**: The initialization of the "Capability Vectors" is quite vague. It’s not clear if the authors are just using a pre-trained embedding model on agent descriptions or using an LLM to extract specific "skill dimensions" (e.g., coding, logic). What is the dimension $d$ of these vectors, and do they carry any interpretable semantics?
- **Vector Dynamics and Reliability**: While the paper mentions that vectors are updated based on task outcomes, the specific mathematical update rules are under-defined. Furthermore, there is no discussion on how to measure or validate the reliability of these updated capability representations over time..
- **Quantifying Query Demand**: The process of quantifying "Capability Demand" via LLM "intent sketches" feels like a "black box". I would like to see more clarity on how the LLM ensures these requirements are accurately and consistently mapped to the same vector space as the agent capabilities.
- **Stale Insights in Dynamic Environments**: In real-world tasks where tools or environments change, historical "Insights" can quickly become obsolete. The paper doesn't explain how the system detects or prunes these outdated structures to prevent the graph from being cluttered with "dead" experience.
- **On Originality and Technical Novelty**: While the application of a unified heterogeneous graph to multi-agent systems is well-motivated, the core technical contribution, combining multiple subgraphs into a single heterogeneous structure, is a relatively standard practice in the broader GNN literature. The novelty here seems to lie more in the application.

### Experimental Concerns

- **Error Analysis**: Given the complexity of the multi-agent interactions, I was hoping to see a deep dive into failure cases. An error analysis would help us understand whether failures stem from incorrect graph retrieval, poor insight distillation, or coordination breakdowns between agents.
- **Statistical Significance**: None of the results in Table 1 or 2 include standard deviations or p-values. Since LLM-based systems are inherently stochastic, reporting results from a single run is risky. Without seeing the variance across multiple trials, it's hard to be fully confident in the reported gains.
- **Scope of "Scalability"**: The paper emphasizes being a "scalable" framework, but the MAS experiments are limited to a configuration of 5 agents. I'm not sure if 5 agents are enough to truly demonstrate the framework's ability to coordinate large-scale agent populations as the title suggests.

---

> ### Author Rebuttal · Authors · 2026-03-29
>
> **Dear Reviewer tKPB,**
>
> Thanks for your careful reading and constructive feedback. We respond to each point below.
>
> > **W1:** Complexity and Scalability Analysis
> >
>
> Thanks for raising this insightful point. SCHEMAS is not built on a fully connected graph: even with many tools, the heterogeneous graph remains sparse in practice, since edges are introduced only through observed interactions rather than full pairwise connectivity. Moreover, low-utility edges are continuously removed to help prevent uncontrolled graph growth. Theoretically, the cost of memory construction is approximately $O(k \cdot |\mathcal{N}_h|)$, where $k$ is the number of retrieved queries and $|\mathcal{N}_h|$ denotes the size of the corresponding \(h\)-hop local trace neighborhood.
>
> > **W2:** Under-specified Capability Vectors & **W3:** Vector Dynamics and Reliability & **W4:** Quantifying Query Demand
> >
>
> We thank the reviewer for this thoughtful comment. Agents and queries are both represented in the same 6-dimensional capability space **(see our response to Reviewer pF4A, W2 & Q3**). The query capability vector is first initialized by the LLM, and its final requirement vector $C_{\text{new}}^{\mathcal Q}$ is then obtained by aggregating information from neighboring retrieved queries. Matching is subsequently performed by comparing $C_{\text{new}}^{\mathcal Q}$ with candidate agents’ capability vectors in the shared space. Suppose there are three agents with capability vectors $C_1^{\mathcal A} = [0.9, 0.2, 0.2, 0.3, 0.2, 0.8]$, $C_2^{\mathcal A} = [0.2, 0.1, 0.3, 0.2, 0.95, 0.2]$, and $C_3^{\mathcal A} = [0.3, 0.8, 0.2, 0.2, 0.3, 0.1]$.  If the  query requirement vector is $C_{\text{new}}^{\mathcal Q} = [0.9, 0, 0, 0, 0, 0.9]$, it indicates that the query mainly requires **reasoning** and **spatial** capabilities. In this case, Agent 1 would be selected. In addition, these capability vectors are updated online in a gradual manner based on execution feedback and task outcomes, allowing each agent’s competence profile over the predefined ability dimensions to evolve with observed behavior.
>
> > **W5:** Stale Insights in Dynamic Environments
> >
>
> Thanks for raising this insightful point. The insights in SCHEMAS are not treated as static rules, but are continuously distilled from new execution traces and updated online as tools or environments change. New insights are also compared with existing ones and merged when sufficiently similar, which helps prevent redundant or repeated insights from accumulating in the graph (as described in the **Insight Graph Update** section).
>
> > **W6:** On Originality and Technical Novelty
> >
>
> We thank the reviewer for this thoughtful comment. To the best of our knowledge, SCHEMAS is the first framework to unify queries, agents, tools, and insights within a single heterogeneous graph, while closing the loop among query-guided retrieval, capability-aware routing, execution, and trace-driven write-back under a shared memory structure. Thus, the novelty lies not in the heterogeneous graph alone, but in how it is used in multi-agent reasoning as a unified structure for memory, coordination, retrieval, and continual updates. We believe this goes beyond a straightforward application, since prior work typically models only isolated components rather than integrating them into a unified, self-evolving, and scalable MAS framework.
>
> > **W7:** Error Analysis
> >
>
> We thank the reviewer for this valuable suggestion. In practice, we observed that one agent successfully identifies a useful intermediate state, but the next agent fails to use it effectively, leading to repeated actions or ineffective loops. This is a challenging issue in multi-agent collaboration. To mitigate it, we incorporate repeated-action detection and re-validate subgoals based on the latest environment state, so that repeated behaviors can trigger correction.
>
> > **W8:** Statistical Significance
> >
>
> Thanks for your valuable suggestion, and we apologize for missing the details! We would like to clarify that the experimental results reported in Tables 1 and 2 are already averaged over three runs, rather than being taken from a single trial.  We will supplement the camera-ready version with the corresponding standard deviations to make the robustness of the reported gains clearer.
>
> > **W9:** Scope of "Scalability"
> >
>
> We thank the reviewer for raising this important point. SCHEMAS adopts a decentralized multi-agent design, where agents coordinate through the shared memory rather than relying on a single central agent, making the framework more suitable for scaling to more agents. Moreover, “scalability” also refers to organizing and retrieving structured memory over growing numbers of queries, tools, and insights within a unified heterogeneous graph. Additional evidence of scalability is provided in our response to **Reviewer CvUD, Q1**.

---

> > ### Author Rebuttal · Reviewer_tKPB · 2026-04-03
> >
> > Thank you for the rebuttal. The authors have addressed my concerns. I will keep my original positive score.

---

> > > ### Author Response · Authors · 2026-04-03
> > >
> > > Dear Reviewer tKPB,
> > >
> > > Thank you for your thoughtful comments and for taking the time to review our rebuttal. We are grateful that our clarifications have addressed your concerns and appreciate your support.
> > >
> > > Warm regards,
> > >
> > > The Authors

---

### Official Review · Reviewer_pF4A · 2026-03-12

**Soundness:** 3
**Presentation:** 3
**Significance:** 3
**Originality:** 2
**Overall Recommendation:** 4
**Confidence:** 4

**Summary:**

This paper presents SCHEMAS, a scalable heterogeneous graph framework designed to organize and optimize multi-agent systems. The graph explicitly models four entity types: query, agent, tool, and distilled insight, along with their typed interdependencies. Upon receiving a new query, the system generates an intent sketch via an LLM to perform meta-path neighborhood filtering and embedding similarity search, constructing a query-conditioned memory subgraph. This subgraph is then used to route the task to the most appropriate agent via capability matching. After execution, a trace-driven update mechanism writes the interaction back to the graph, continually refining agent capabilities, tool statistics, and relational weights. The framework is evaluated across 8 tool-use and interactive benchmarks, demonstrating consistent performance improvements over existing memory-augmented MAS baselines.

**Compliance With Llm Reviewing Policy:**

Affirmed.

**Final Justification:**

The author's response has addressed my concerns. I maintain my positive scores.

**Key Questions For Authors:**

- The method uses an LLM to generate an intent sketch before memory retrieval. Which exact model is used for this step in your experiments? Is it always the identical backbone as the evaluated system, or does it rely on a stronger external model (like GPT-4o)? If the latter, how do you justify the fairness of the baseline comparisons?
- Could the authors provide a wall-clock latency analysis of SCHEMAS, specifically isolating the overhead of the intent-sketch generation and meta-path-based retrieval, in addition to the token cost?
- How exactly are the query requirement vector ($C_{new}^Q$) and the agent capability vectors ($C_i^A$) initialized and calculated in practice? The current paper leaves these representations and update functions (e.g., Aggregate, Calibrate, Coop) rather abstract. Please provide concrete implementation details.
- Since the heterogeneous graph continually grows over time with new execution traces, does the system implement any memory-budget control, graph pruning, node eviction, or forgetting mechanisms to prevent retrieval degradation and graph bloat in long-running settings?
- In the main result tables (Table 1 and 2), are all methods compared under strictly matched backbone settings? If the results for MAS baselines/SCHEMAS are averaged or selected across multiple backbones, please explicitly clarify how this aggregation is done.

**Limitations:**

Yes

**Strengths And Weaknesses:**

### Strength:
- Formalizing the disparate components of an MAS (queries, tools, agents, and insights) into a single heterogeneous graph is a highly natural and practically useful abstraction for long-horizon, tool-augmented MAS.
- The framework is reasonably complete at the systems level. It elegantly combines query-conditioned retrieval, capability-aware task routing, and trace-driven write-back updates into a closed, self-evolving loop.
- The empirical evaluation is extensive, covering both general tool-use benchmarks and memory-intensive interactive tasks. The included ablations on meta-path retrieval and dynamic task allocation help substantiate the claimed gains.
- The reported token-cost analysis suggests that the method achieves improved performance without an obviously excessive token budget, which is highly relevant for practical deployment.
### Weakness:
- My major concern is the reliance on an LLM-generated "intent sketch" prior to memory retrieval. The paper mentions using "e.g., GPT-4o" for this step. If a powerful external model like GPT-4o is used to route and plan the graph retrieval while the evaluated system/baselines rely on smaller open-source models (e.g., Qwen-2.5), this introduces a severe unfair advantage. Furthermore, adding an LLM inference stage purely for retrieval introduces latency and computational overheads that are not carefully analyzed.
- Several core modules are presented mostly as high-level operators or black-box functions (e.g., Aggregate, Calibrate, Coop, Trans, Distill, Merge, and capability update rules). Specifically, the mathematical definition and initialization of the query requirement vector ($C_{new}^Q$) and agent capability vectors ($C_i^A$) are vague. As a result, the method is less reproducible and less technically grounded than its mathematical formalism implies.
- A large part of the contribution comes from integrating existing ideas: graph memory, retrieval-augmented execution, capability-based routing, and trace write-back, into a heterogeneous graph. This represents strong systems engineering rather than a fundamentally new algorithmic paradigm.
- As query nodes, similarity edges, and insight nodes continuously accumulate, the framework will inevitably face graph bloat, retrieval noise, and memory-quality degradation over time. The paper does not clearly discuss pruning, forgetting, or memory-budget control mechanisms for long-term operation.
- The experimental setup describes multiple LLM backbones for MAS baselines, but the main result tables do not make it sufficiently transparent whether all compared methods are evaluated under exactly matched backbone conditions.

---

> ### Author Rebuttal · Authors · 2026-03-29
>
> **Dear Reviewer pF4A,**
>
> Thank you for your careful reading and constructive feedback. We address your concerns below.
>
> > **W1:** My major concern is ... &
> **Q1:** The method uses an  ...
> >
>
> Thanks for raising this insightful point. We apologize for the confusing wording in the manuscript. The phrase “e.g., GPT-4o” was only an illustrative example and may be misleading. In all experiments, intent-sketch generation uses **Qwen-2.5-32B**, not a stronger external model. This holds for both the single-agent and multi-agent settings. As shown below, removing the sketch leads to consistent performance drops. This suggests that the gain comes from the intent-sketch mechanism itself rather than from using a stronger external model, while also indicating that the additional retrieval stage provides meaningful guidance for execution. A detailed analysis of the additional computational overhead is provided in our response to Q2.
>
> |Method|intent-sketch| ALFWorld(%) | SciWorld(%) |
> | --- | --- | --- | --- |
> | SCHEMAS | ✓ | **60.17** | **36.78** |
> |  | ✗ | 58.33 | 34.27 |
>
> > **Q2:** Could the authors provide a wall-clock latency analysis ...
> >
>
> Thanks for the great question. We have measured the average runtime on the **ALFWorld** benchmark over 100 randomly sampled tasks and isolated the overhead introduced by intent-sketch generation and meta-path-based retrieval. As shown below, the results indicate that both intent-sketch generation and meta-path-based retrieval introduce only modest latency overhead, while also providing significant gains in downstream performance.
>
> | Method | Execution Time(s) | Sketch Time(s) | Meta-path Time(s) | Performance(%) |
> | --- | --- | --- | --- | --- |
> | SCHEMAS | 125.27 | 13.20 | 7.10 | 60.17 |
> | w/o Sketch | 114.86 | 0 | 8.90 | 58.33 |
> | w/o Meta-path | 120.54 | 14.80 | 0 | 57.21 |
>
> > **W2:** Several core modules ... &
> **Q3:**  How exactly are the query ...
> >
>
> We thank the reviewer for this important comment. The query requirement vector $C_{new}^{\mathcal Q}$ is intended as a semantic representation of the capability requirements induced by the input query, while the agent capability vector $C_i^{\mathcal A}$ represents the competence profile of agent $\mathcal A$ in a predefined ability space for task assignment and routing. As illustrated below, we define an explicit ability space, where each agent is initialized with capability values of 0.6 on all ability dimensions. In addition, tasks in each benchmark are associated with corresponding task-to-ability mappings, which are used to support agent-task matching.  More details on query capability mapping and calculation in the shared ability space are provided in our response to **Reviewer tKPB (W2–W4)**. Additionally, the operators such as Aggregate, Calibrate, Coop, and Trans are explicitly defined and instantiated in the current system through concrete retrieval, similarity, and update procedures, with each operator corresponding to a specific functional step in the overall pipeline. We will make all implementation details explicit in the camera-ready version.
>
> | Task | Abilities |
> | --- | --- |
> | formal_fallacies | reasoning |
> | object_counting | mathematical, spatial |
> | word_sorting | mathematical |
> | dyck_languages | mathematical, language |
>
> | Ability | Value |
> | --- | --- |
> | reasoning | 0.6 |
> | mathematical | 0.6 |
> | language | 0.6 |
> | knowledge | 0.6 |
> | sequence | 0.6 |
> | spatial | 0.6 |
>
> > **W4:** As the graph continuously grows ... &
> **Q4:** Since the graph continually grows ...
> >
>
> We thank the reviewer for raising this important question. Our current framework includes partial controls rather than operating in a completely unconstrained manner. Specifically, insight merging reduces redundant memory accumulation, while the system continuously updates usage-related statistics for queries and tools, such as invocation frequency and success rate, to help indicate which memories remain useful over time. In addition, SCHEMAS performs query-guided subgraph retrieval instead of reasoning over the full graph at execution time, which limits the effective memory footprint for each query. We acknowledge that the current framework does not yet fully address long-term memory management, and we consider this an important direction for future work.
>
> > **W5:** The experimental setup describes ... &
> **Q5:** In the main result tables (Table 1 and 2) ...
> >
>
> We thank the reviewer for raising this important question. We would like to clarify that the results in Tables 1 and 2 are **not averaged or selected across multiple backbones**. Instead, all methods are evaluated under fixed and matched backbone settings within each scenario. Specifically, in the single-agent setting, all compared methods use Qwen-2.5-32B. In the multi-agent setting, all MAS baselines and SCHEMAS are instantiated under the same multi-agent framework and the same five-agent backbone configuration: Qwen-2.5-7B, Qwen-2.5-14B, Qwen-2.5-32B, QwQ-32B, and GPT-4o-mini.

---

> > ### Author Rebuttal · Reviewer_pF4A · 2026-04-01
> >
> > Thanks for the rebuttal. All my concerns are addressed. I will keep my original positive scores and have updated my confidence to 4.

---

> > > ### Author Response · Authors · 2026-04-01
> > >
> > > Thank you, Reviewer pF4A, for your positive evaluation and kind response. Your support and increased confidence mean a lot to us. Once again, we sincerely appreciate your thoughtful and constructive feedback throughout the review process.

---

### Official Review · Reviewer_ikGi · 2026-03-13

**Soundness:** 2
**Presentation:** 2
**Significance:** 3
**Originality:** 2
**Overall Recommendation:** 3
**Confidence:** 3

**Summary:**

This paper provide a heterogeneous graph framework, named SCHEMAS. It unifies four entity types: query, agent, tool, and distilled insight to enable query-guided reasoning and continual evolution under the multi-agent settings. Extensive experiments demonstrate that the proposed method can balance performance and token cost across diverse benchmarks.

**Compliance With Llm Reviewing Policy:**

Affirmed.

**Final Justification:**

It address my main concerns. I will raise my score to 3.

**Key Questions For Authors:**

See above

**Strengths And Weaknesses:**

Strength:
1. Good visualization.
2. This method achieves good performance while keeping the token usage reletively low.

Weakness:
1. The authors argue that "Scalable Heterogeneous Graph" is one of the major contribution. However, this advantage is mainly a basic property of heterogeneous graph, rather than the specific proposed framework.
2. The authors mention various kernelized similarity functions in section 4. However, there is no relevant ablation study.
3. The proposed method relies heavily on hyper-params tuning, which is not analyzed across the paper.
4. I think this method needs a good cold start phase. However, I cannot see any analysis. How does the performance evolve as more task are finished?

---

> ### Author Rebuttal · Authors · 2026-03-29
>
> **Dear Reviewer ikGi,**
>
> Thanks for your valuable feedback. We address your concerns as follows.
>
> > **W1:** The authors argue that "Scalable Heterogeneous Graph" is one of the major contribution.
> >
>
> We thank the reviewer for this thoughtful comment. To the best of our knowledge, SCHEMAS is the first framework to unify structured memory and coordination in MAS through a shared heterogeneous graph of queries, agents, tools, and insights. We agree that scalability is an inherent property of heterogeneous graphs. However, the contribution of SCHEMAS lies not in graph-level scalability itself, but in using the graph to support system-level scalability in MAS through query-guided retrieval, capability-aware routing, and continual updates—thereby enabling memory reuse, agent coordination, and continual learning. Empirically, SCHEMAS consistently achieves strong performance across diverse benchmarks while maintaining favorable token efficiency. As **Reviewer tKPB** noted, “The paper does a great job of unifying queries, agents, tools, and insights into a single, **extensible** heterogeneous graph schema.” **Reviewer CvUD** pointed out that “The four-entity-type heterogeneous graph framework provides a principled approach to modeling complex MAS interactions, **going beyond existing agent-centric or tool-centric representations**,” and that it “addresses fundamental **scalability bottlenecks** in existing MAS frameworks.”
>
> > **W2:** The authors mention various kernelized similarity functions in section 4. However, there is no relevant ablation study.
> >
>
> Thanks for your valuable suggestion. All experiments reported in the paper use the **Cosine kernel** as the default setting. In our preliminary experiments, we compared Cosine, RBF, and Gaussian kernels on ALFWorld and SciWorld, and found only limited performance differences, indicating that SCHEMAS is not highly sensitive to the exact kernel choice. We selected the Cosine kernel because it showed lower variance and required no additional hyperparameter tuning. Since this comparison had only a minor impact on the overall conclusions and space was limited, we did not include it in the main paper, but we will add it in the camera-ready version.
>
> | Similarity Function | ALFWorld(%) | SciWorld(%) |
> | --- | --- | --- |
> | Cosine kernel | 60.17 ± 0.21 | 36.78 ± 0.34 |
> | RBF kernel | 60.08 ± 1.21 | 36.81 ± 1.05 |
> | Gaussian kernel | 60.22 ± 1.17 | 36.66 ± 1.43 |
>
> > **W3:** The proposed method relies heavily on hyper-params tuning, which is not analyzed across the paper.
> >
>
> Thanks for raising this insightful point. We would like to clarify that SCHEMAS does not rely heavily on tuning a large number of hyperparameters. In our framework, the two main hyperparameters are the retrieval hop number ($h$) and top-$k$. The results suggest that SCHEMAS is reasonably stable within a moderate parameter range, while overly large retrieval ranges can introduce noisier memory and degrade performance. Analyses of other hyperparameters are provided in our response to **Reviewer CvUD (W1)**. We will include these sensitivity analysis in the camera-ready version.
>
> | h-hop  | ALFWorld(%) | SciWorld(%) |
> | --- | --- | --- |
> | 1-hop | **60.19** | 34.29 |
> | 2-hop | 55.34 | **36.78** |
> | 3-hop | 54.22 | 33.26 |
>
> | top-k | ALFWorld(%) | SciWorld(%) |
> | --- | --- | --- |
> | 1 | 58.23 | 34.01 |
> | 3 | 58.86 | 34.72 |
> | 5 | **60.28** | 35.01 |
> | 7 | 56.45 | **36.82** |
> | 9 | 54.11 | 29.35 |
>
> > **W4:**  I think this method needs a good cold start phase. However, I cannot see any analysis. How does the performance evolve as more task are finished?
> >
>
> We thank the reviewer for raising this important point. We would like to clarify that SCHEMAS does **not need a “good” cold-start phase**. Instead, SCHEMAS starts from a simple initialization: the agent graph is initialized as a fully connected graph, with agent capability profiles initialized in the predefined ability space (**see our response to Reviewer pF4A, W2 & Q3**), the tool graph is built from the available tool inventory and static tool descriptions (**see Tool Graph Update section, Line 320** ), and the query graph is initialized based on semantic similarity between queries. Only the insight graph is absent at initialization, and it begins to form after the first completed task. To further address the reviewer’s concern, we analyze how performance evolves on **ALFWorld** as more tasks are completed. The results show that SCHEMAS benefits progressively from accumulated experience, with performance improving steadily in the early and middle stages and gradually saturating in the later stage. We will clarify this cold-start setting and include the growth-curve analysis in the camera-ready version.
>
> | Finished Tasks | Success(%) |
> | --- | --- |
> | 10 | 50.24 |
> | 20 | 51.37  |
> | 40 | 54.86 |
> | 60 | 56.94 |
> | 80 | 58.21 |
> | 100 | 59.34 |
> | 120 | 60.17 |
> | 130 | 60.22 |

---

> > ### Author Rebuttal · Reviewer_ikGi · 2026-04-04
> >
> > Thanks for your reply. I will update my score.

---

> > > ### Author Response · Authors · 2026-04-06
> > >
> > > Dear Reviewer ikGi,
> > >
> > > Thank you again for your thoughtful review and for revisiting and updating your assessment after reading our rebuttal. We sincerely appreciate your time and consideration, and we are glad that our response addressed your concerns.
> > >
> > > We are also grateful for your constructive suggestions, which have been helpful in further clarifying the paper. We will carefully take your feedback into account in the revised version.
> > >
> > > Thank you again for your careful evaluation and valuable comments.
> > >
> > > Warm regards,
> > >
> > > The Authors

---

### Official Review · Reviewer_CvUD · 2026-03-13

**Soundness:** 3
**Presentation:** 3
**Significance:** 3
**Originality:** 2
**Overall Recommendation:** 4
**Confidence:** 4

**Summary:**

This paper presents **SCHEMAS**, a scalable heterogeneous graph framework aimed at improving query-guided reasoning in multi-agent systems (MAS). It tackles key challenges in coordinating complex interactions among queries, agents, tools, and insights within MAS environments. The main contribution is a unified graph representation that explicitly captures four types of entities and their typed relationships, together with a query-driven structured memory mechanism that builds relevant subgraphs for efficient retrieval and decision-making. The framework also incorporates dynamic update processes that support continual self-improvement through trace-based updates to node attributes, edge weights, and graph structure. Experiments conducted on eight benchmarks show consistent gains over existing baselines, with especially strong performance on tool-use tasks and in interactive scenarios.

**Compliance With Llm Reviewing Policy:**

Affirmed.

**Final Justification:**

The rebuttal addresses most of my concerns, and the additional empirical results demonstrate the robustness of the proposed approach. Accordingly, I maintain my positive score and increase my confidence score.

**Key Questions For Authors:**

1. Specifically, how does its performance change as the number of tools, agents, or historical queries grows beyond the sizes used in the current benchmarks? For example, if the framework were tested on long-horizon reasoning tasks such as those in the HLE benchmark, would the approach remain stable in terms of efficiency and accuracy as the interaction history and graph size expand?

**Limitations:**

Yes

**Strengths And Weaknesses:**

## Strengths

* **Technical novelty and innovation**: The four-entity-type heterogeneous graph framework (Query-Agent-Tool-Insight) provides a principled approach to modeling complex MAS interactions, going beyond existing agent-centric or tool-centric representations. The query-guided memory construction mechanism elegantly addresses scalability through meta-path-based subgraph extraction, while the dynamic co-evolution framework enables genuine continual learning through trace-driven updates.

* **Experimental rigor and validation**: The evaluation spans both tool-use tasks (ToolBench, API-Bank, ToolHop, TMDB) and interactive tasks (ALFWorld, ScienceWorld, PDDL, HotpotQA), providing broad validation across different reasoning paradigms. The paper includes fair baseline comparisons with both single-agent and multi-agent approaches, proper ablation studies distinguishing component contributions, and scalability analysis demonstrating favorable token efficiency.

* **Clarity of presentation**: The methodology follows a logical progression from preliminary definitions through memory construction to dynamic updates. Mathematical formulations are precisely specified with clear heterogeneous graph definitions and meta-path templates. Visualizations effectively illustrate the complete pipeline and graph structure.

* **Significance of contributions**: The framework addresses fundamental scalability bottlenecks in existing MAS frameworks, enables continual adaptation through experience-based learning, and provides broad applicability through modular design compatible with different LLM backbones and tool ecosystems.

## Weaknesses

* **Technical limitations or concerns**: The paper lacks convergence guarantees or stability analysis for dynamic update mechanisms - exponential moving averages and threshold-based merging could lead to instability or information loss. The choice of three meta-path templates appears somewhat arbitrary without principled design guidelines. The capability vector representation doesn't adequately handle evolving tool capabilities or newly introduced tools. Memory management lacks discussion of bounds, garbage collection, or handling outdated information.

* **Clarity or presentation issues**: Mathematical notation has inconsistencies (distinction between G(Q_new) and G̃(Q_new)). Implementation details about similarity functions, threshold selection, and computational complexity are insufficiently specified. Some figure quality could be improved with larger text and clearer visual distinctions.

---

> ### Author Rebuttal · Authors · 2026-03-29
>
> **Dear Reviewer CvUD,**
>
> We would like to thank you for your careful reading and constructive suggestions. We respond to each point below.
>
> > **W1:** The paper lacks ...
> >
>
> We thank the reviewer for this important point. We have conducted a sensitivity analysis on the EMA coefficient $\beta$. For the merge threshold, we use a fixed value of 0.8 in all experiments. Based on our preliminary tests, the performance is not highly sensitive to moderate variations around this value, so we keep it fixed. In addition, we also analyze two more sensitive hyper-parameters, namely $h$-hop and top-$k$, as well as a growth-curve analysis of performance over time (see our response to **Reviewer ikGi, W3 & W4**). Together, these results provide empirical evidence that the framework remains stable under dynamic updates.
>
> |EMA  ($\beta$)|ALFWorld(%)|
> | --- | --- |
> | 0.50 | 58.37 |
> | 0.70 | 59.88 |
> | 0.80 | 60.17 |
> | 0.90 | 60.11 |
> | 0.95 | 60.12 |
>
> > **W2:** The choice of three meta-path templates ...
> >
>
> We thank the reviewer for raising this point. We would like to clarify that these meta-paths were chosen as a minimal set of short, schema-driven paths aligned with the heterogeneous graph structure. Specifically, Query–Tool–Query captures shared tool-use patterns, Query–Agent–Query captures shared capability and executor relevance, and Query–Insight–Query captures shared reasoning strategies. Together, they cover the three main information channels in our framework while avoiding longer meta-paths that are typically noisier. We do not claim that this set is exhaustive or uniquely optimal, but rather that it is a simple and effective design guided by the graph schema.
>
> > **W3:** The capability vector ...
> >
>
> We thank the reviewer for pointing this out. In our framework, capability vectors are defined only for queries and agents, not for tools. Tool-related information is modeled separately in the tool graph through tool nodes, relations, and execution statistics. Therefore, newly introduced tools are handled primarily through **tool-graph updates**, not through the capability vectors. Once added, a new tool can be retrieved through the memory subgraph and considered by the assigned agent during task execution. Meanwhile, capability vectors are dynamically updated based on execution feedback, enabling agents to better utilize newly introduced tools over time.
>
> > **W4:** Memory management ...
> >
>
> We thank the reviewer for raising this important point. Our current framework includes partial memory management mechanisms, such as insight merging to reduce redundancy and query-guided subgraph retrieval to limit the effective memory footprint during execution. However, we do not yet include explicit memory bounds, or a systematic strategy for outdated information. We will clarify this limitation in the revision and discuss it as an important direction for future work.
>
> > **W5:** Mathematical notation ...
> >
>
> We thank the reviewer for the helpful comment. In our formulation, we do not explicitly define $G(Q_{\text{new}})$ or $\tilde{G}(Q_{\text{new}})$ in the paper. Specifically, $\tilde{Q}\_\{\text{new}}$ denotes the retrieved query set, $G_Q^{(h)}$ denotes the $h$-hop trace subgraph of each retrieved query, and $\bar{G}(Q\_\{\text{new}})$ denotes the final memory subgraph obtained by their union. We will unify and clarify the notation in the camera-ready version.
>
> > **W6:** Implementation details ...
> >
>
> Thanks for your valuable suggestion, and we apologize for the missing details. We have clarified the similarity functions in our response to **Reviewer ikGi ,W2**, and the threshold selection and other implementation details in **W1**.
>
> > **Q1.** Specifically, how does its performance change...
> >
>
> We thank the reviewer for this important question. We have evaluated SCHEMAS on the **HLE** benchmark [1]. Following the experimental setup of DeepAgent-32B-Base[2], SCHEMAS achieves the best **Pass@1** across the text-only (Text), multimodal (MM), and full-benchmark (All), surpassing all compared baselines, including WebThinker[3] and HiRA[4]. These results suggest that the framework remains effective on **more challenging long-horizon tasks** and provide preliminary evidence of its scalability to larger numbers of tools, agents, and historical queries.
>
> |Method|Text(%)|MM(%)|All(%)|
> |---|---|---|---|
> |WebThinker| 14.1 | 9.2 | 13.3|
>  HiRA | 14.3| 10.8 | 13.5 |
> |DeepAgent-32B-Base| 19.3 | 13.1 | 17.9 |
> |**SCHEMAS (ours)**| **22.2** | **16.4** | **20.5** |
>
> *[1] Phan L, Gatti A, Han Z, et al. Humanity's last exam. arXiv:2501.14249, 2025.*
>
> *[2] Li X, Jiao W, Jin J, et al. Deepagent: A general reasoning agent with scalable toolsets. arXiv:2510.21618, 2025.*
>
> *[3] Li X, Jin J, Dong G, et al. Webthinker: Empowering large reasoning models with deep research capability. arXiv:2504.21776, 2025.*
>
> *[4] Jin J, Li X, Dong G, et al. Decoupled planning and execution: A hierarchical reasoning framework for deep search. arXiv: 2507.02652, 2025.*

---

> > ### Author Rebuttal · Reviewer_CvUD · 2026-04-02
> >
> > I appreciate the authors’ response and the additional results on challenging long-horizon tasks, which further demonstrate the robustness over different task complexities. My concerns have been addressed; I will maintain my positive score and increase my confidence from 2 to 4.

---

> > > ### Author Response · Authors · 2026-04-03
> > >
> > > Dear Reviewer CvUD,
> > >
> > > We are deeply grateful for your positive evaluation and for the time and effort you have invested in reviewing our work and rebuttal. We truly appreciate your recognition of the additional results on challenging long-horizon tasks and your acknowledgment of the robustness of our approach across different task complexities. Your continued support and increased confidence are truly encouraging to us, and we will further improve the revised version by carefully incorporating your valuable feedback.
> > >
> > > Thank you once again for your thoughtful and constructive feedback.
> > >
> > > Warmest regards,
> > > The Authors

---

### Decision · Program_Chairs · 2026-04-30

**Decision:**

Reject

**Comment:**

This paper introduces SCHEMAS, a framework that models multi-agent systems (MAS) as a scalable heterogeneous graph unifying four core entity types: queries, agents, tools, and distilled insights. To coordinate complex interactions, the system employs a query-driven structured memory mechanism that retrieves relevant subgraphs for capability-aware task routing. Following execution, a dynamic trace-driven write-back mechanism updates the graph's node attributes and relational weights, enabling continuous self-evolution and experience reuse.

Summary Of Weaknesses and Suggested Revisions:
- Though the authors attempted to frame their contribution as a novel systems-level integration of a self-evolving multi-agent application, the approach ultimately relies on combining existing concepts rather than proposing fundamentally new algorithmic paradigm.
- Multiple reviewers expressed concern over unbounded graph growth, stale insights, and retrieval noise over time. While the authors pointed to existing partial controls like insight merging and sub-graph footprint limits, the lack of explicit garbage collection or memory-budget pruning mechanisms are key limitations to be addressed.

Overall, the paper can benefit from more exciting and novel technical contribution points, as well as more solid empirical validation, before future acceptance.